# Gen2Seg: Generative Models Enable Generalizable Instance Segmentation

**Om Khangaonkar, Hamed Pirsiavash**
University of California, Davis
reachomk.github.io/gen2seg

## Abstract

By pretraining to synthesize coherent images from perturbed inputs, generative models inherently learn to understand object boundaries and scene compositions. How can we repurpose these generative representations for general-purpose perceptual organization? We finetune Stable Diffusion and MAE (encoder+decoder) for category-agnostic instance segmentation using our instance coloring loss exclusively on a narrow set of object types (indoor furnishings and cars). Surprisingly, our models exhibit strong zero-shot generalization, accurately segmenting objects of types and styles unseen in finetuning. This holds even for MAE, which is pretrained on unlabeled ImageNet-1K only. When evaluated on unseen object types and styles, our best-performing models closely approach the heavily supervised SAM, and outperform it when segmenting fine structures and ambiguous boundaries. In contrast, existing promptable segmentation architectures or discriminatively pretrained models fail to generalize. This suggests that generative models learn an inherent grouping mechanism that transfers across categories and domains, even without internet-scale pretraining. Please see our website for additional qualitative figures, code, and a demo.

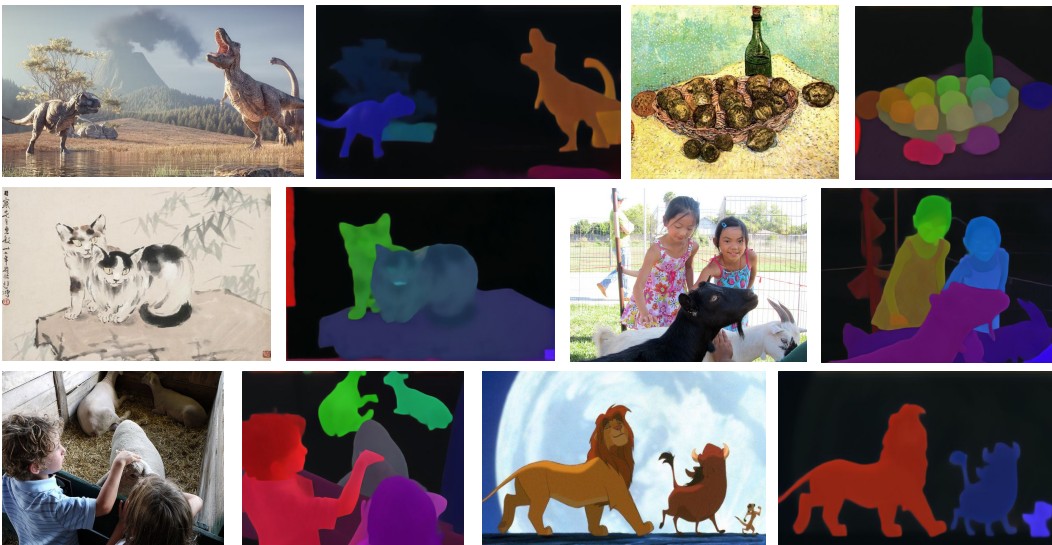

Figure 1: **The model that generated the segmentation maps above has never seen masks of humans, animals, or anything remotely similar.** We fine-tune generative models for instance segmentation using a synthetic dataset that contains *only* labeled masks of indoor furnishings and cars. Despite never seeing masks for many object types and image styles present in the visual world, our models are able to generalize effectively. They also learn to accurately segment fine details, occluded objects, and ambiguous boundaries.

# 1 INTRODUCTION

Humans learn to carve the visual world into discrete, persistent objects from limited experience. A toddler who has mostly handled cups, chairs, and toys at home can still recognize the zebra, giraffe, and lion on their first trip to the zoo as distinct objects, even without knowing what they are called. This ability extends to abstract depictions: people can understand line drawings and artworks *even without prior exposure to them* (Hochberg & Brooks, 1962). These observations suggest human vision acquires general, transferable mechanisms for grouping pixels into objects.

Modern vision systems often show "zero-shot" transfer to new datasets without domain-specific fine-tuning, but typically rely on broad labeled datasets that attempt to cover many object categories and styles. We ask a different question through the lens of instance segmentation: can a model learn from only a very narrow slice of the visual world and still generalize to unseen object types and image styles? We focus on a stricter "zero-shot" setting, where we explore how the model performs on object types it has never seen a mask for.

We hypothesize that generative models are particularly well posed to succeed at this task. Because they learn to synthesize scenes from minimal cues (e.g., a text prompt or a corrupted image), they must learn to implicitly represent the parts that make up the image. This allows generative models to produce scene compositions unlike anything seen in pretraining data, such as "a Van-Gogh style painting of a panda driving a car."

We introduce a simple finetuning method that taps into these generative priors. We first experiment with a Masked Autoencoder (MAE; encoder+decoder) pretrained on ImageNet-1K only end-to-end using *only* mask supervision from two narrow synthetic domains (indoor furnishings and cars) to generate object instance groupings. Despite this limited supervision, our model is able to generalize to new object types unseen in finetuning, such as people and animals, or novel image styles, such as impressionist art and x-rays. When we experiment with limiting the diversity or complexity of our finetuning dataset, this generalization persists, suggesting it is due to the generative prior.

To explore the effects of internet-scale pretraining, we apply our finetuning method to Stable Diffusion 2. Without seeing *any* labeled masks from the evaluation categories, our finetuned Stable Diffusion achieves performance comparable to SAM across five datasets of distinct domains. Beyond generalization, our models consistently produce crisper boundaries than SAM (e.g., on BSDS500), a behavior that persists even when finetuned on datasets with polygonal edges (e.g., COCO). They also excel at segmenting fine structures (e.g., on iShape) and exhibit object-part compositionality, despite never receiving part-level supervision. We hypothesize these behaviors arise directly from generative pretraining, which must model fine edges, delicate structures, and part–whole relationships to synthesize detailed scenes.

Together, our findings argue that generative pretraining encodes an inherent grouping mechanism that extends beyond both the object types and the image styles seen during finetuning. We hope utilizing the generative representations learned from image synthesis can pave the way for more generalizable and human-like perception, enabling advances in fields where detailed scene understanding is critical, such as robotics, medical imaging, and autonomous systems.

# 2 RELATED WORK

## 2.1 GENERATIVE MODELS FOR PERCEPTION

Generative models, originally developed for image synthesis, have increasingly been adapted for perception tasks in computer vision. A longstanding viewpoint in the field (dating back to Hinton's early work) (Hinton, 2007) posits that learning to *generate* data can aid in *recognizing* it. Early work on GANs (Goodfellow et al., 2014) evaluated whether representations learned by generating images (Radford et al., 2015) or videos (Vondrick et al., 2016) transferred well to image classification or action recognition, respectively, but performance was always far below discriminatively pretrained models. Some works utilized inpainting (Pathak et al., 2016) and colorization (Larsson et al., 2017; Zhang et al., 2016; 2017) as pretext tasks for representation learning, but these were subsequently surpassed by discriminative pretraining (Noroozi et al., 2017; Gidaris et al., 2018; He et al., 2020).

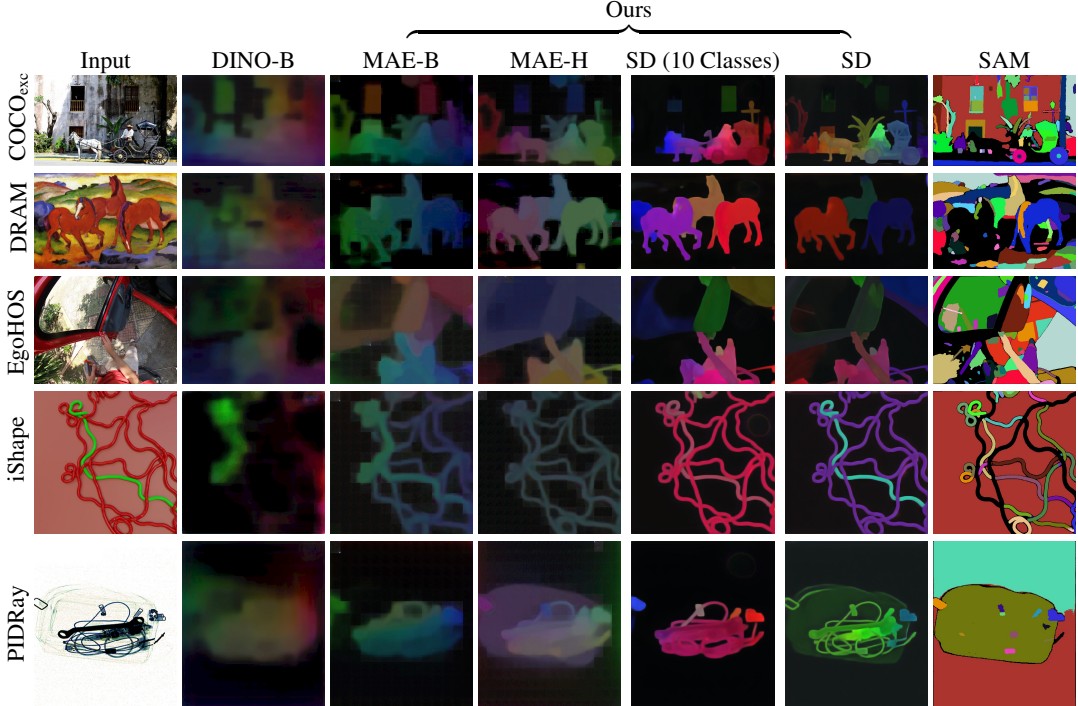

Figure 2: To showcase the potential of generative models for instance segmentation, we highlight an example from each evaluation dataset where most or all of our models outperform SAM, despite never having seen masks of these object types. *SAM often fails on fine structures (wires) or ambiguous boundaries (horses & carriage), leaving black regions where no object was detected.* DINO-B also performs poorly, suggesting that generative pretraining (e.g., MAE, Stable Diffusion) learns strong priors for perceptual grouping.

Recent advancements have demonstrated the efficacy of diffusion models (Sohl-Dickstein et al., 2015; Ho et al., 2020) in various visual tasks. A key advantage of recent diffusion models is the sheer scale of their pretraining; learning from over 2 billion images (Schuhmann et al., 2022) has the potential to outscale existing discriminatively trained models. Their large-scale generative pretraining has since been transferred to many perceptual tasks, such as 3D reconstruction (Liu et al., 2023b; Poole et al., 2022; Wang et al., 2023a), semantic (Baranchuk et al., 2021; Li et al., 2023b; Kawano & Aoki, 2024; Tian et al., 2024) and amodal segmentation (Ozguroglu et al., 2024; Chen et al., 2024), monocular depth (Ke et al., 2024; Zhao et al., 2023), surface normals (Fu et al., 2024), optical flow (Ravishankar et al., 2024), correspondence (Tang et al., 2023), and classification (Li et al., 2023a). Other works have shown that depth, normals, albedo, and segmentation can emerge (albeit with low quality) from generative models (Bhattad et al., 2023; Namekata et al., 2024; Karmann & Urfalioglu, 2024) without finetuning, suggesting similar representations may emerge from the data alone (Dravid et al., 2023; Huh et al., 2024). While there are some prior works that finetune diffusion models for instance segmentation (Fan et al., 2024; Zhao et al., 2025), these works focus on building competitive instance segmenters using large scale data, while we explore generalization through the lens of instance segmentation.

A parallel line of work explores representation learning through Masked Autoencoders (MAE) (He et al., 2022), which achieve state-of-the-art performance across many visual tasks by pretraining to reconstruct masked image tokens before fine-tuning on discriminative objectives. A common practice, however, is to discard the decoder despite it containing rich pixel-level generative features prior to fine-tuning. Recent work (Bar et al., 2022) has demonstrated the encoder and decoder's joint capacity to generalize through visual prompting, generating masks without explicit supervision. In contrast, our approach end-to-end fine-tunes the encoder and decoder and shows that it can generalize to objects whose masks were not observed during fine-tuning.

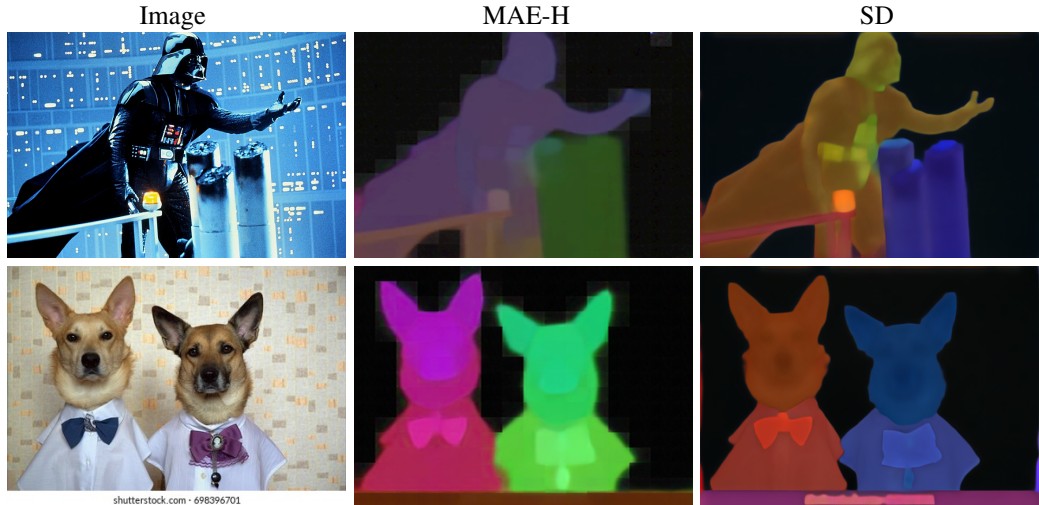

Figure 3: Our models assign similar colors to compositionally related parts of a scene. Vader's mask and body (top), or the bowties and shirts (bottom) are separated by subtly different hues, while distinct colors partition unrelated parts such as his leg and the poles (top), or the dogs and text (bottom). This emerges without any part-level supervision, suggesting generative models learn hierarchical scene representations. See Figures 15 to 21 and Table 7 for more examples and results.

## 2.2 INSTANCE SEGMENTATION

Recent advancements have introduced category-agnostic instance segmentation, enabling models to segment objects without prior knowledge of their classes. The most well known of these is SAM (Kirillov et al., 2023), which learns zero-shot promptable segmentation by finetuning an MAE backbone and mask decoder on the massive SA-1B dataset. SAM and its recent successor SAM2 (Ravi et al., 2024) represent a breakthrough in obtaining general, category-agnostic masks without per-dataset training. However, unlike SAM, which was trained on a massive labeled dataset, our approach aims to leverage generative knowledge to achieve broad instance segmentation. Furthermore, our strongest model is trained for only 29 hours on four RTX6000 Ada (48GB) GPUs on less than 87,000 images and 3.7 million masks of only select categories, while SAM was trained for 68 hours on 256 A100 (80GB) GPUs on 11 million images and 1.1 billion masks of many aspects of the visual world.

Recent works (Wang et al., 2024; 2023b; 2022) train instance segmenters from pseudo-masks automatically derived from contrastive self-supervised features via normalized cuts (Shi & Malik, 2000), or similarity and thresholding. While both our work and theirs use less labelled data compared to prior works (Kirillov et al., 2023), our goals differ. They leverage vast, diverse unlabeled images to synthesize pseudo-labels for downstream finetuning, to reduce reliance on manually annotated labels. In contrast, we use manually annotated labels, but deliberately restrict annotation diversity, to build models that generalize to many novel object types when finetuned on very few. Additionally, the noisy pseudo-labels require iterative retraining, which takes 2–15x longer than our single-pass fine-tuning. Many of the findings from our work, such as our zero-shot generalization and our robustness to noisy labels, provide a path to eventually combine these two directions in future work.

## 3 METHOD

We aim to adapt pretrained generative models, such as Stable Diffusion or MAE, to perform category agnostic instance segmentation. Instance segmentation, in particular, is a *pixel level task*, at which generative models are naturally primed to excel. Specifically, we hypothesize they inherently learn to understand object boundaries and groupings because they must synthesize the objects' core structure and boundaries themselves. In contrast, most SOTA models, such as SAM (Kirillov et al., 2023), extract features using an encoder that discards low-level details. As a result, they must learn mask predictors or feature pyramid networks from scratch to gradually upsample these features to higher resolutions (Lin et al., 2017; Li et al., 2022).

*a) using diffusion model backbone*

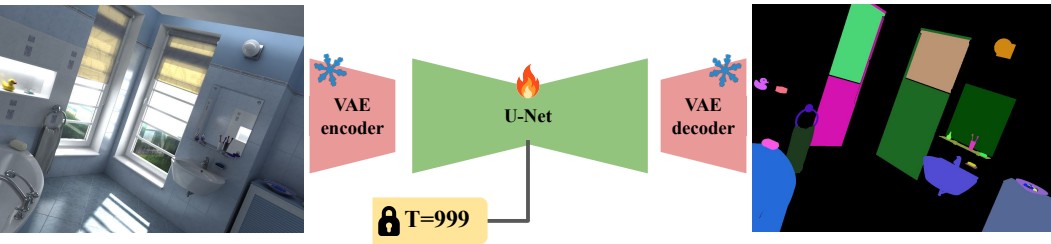

*b) using ImageNet-pretrained MAE backbone*

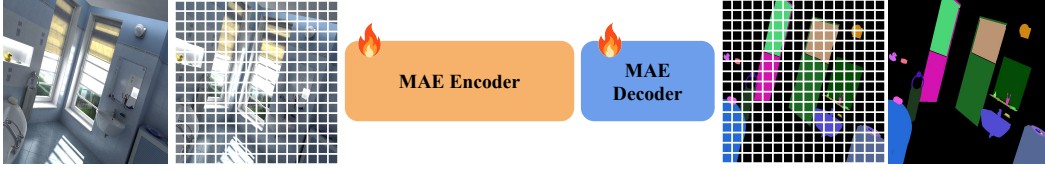

Figure 4: We treat instance segmentation as image-to-image translation problem to **repurpose the generative prior** without task-specific heads. Given an input image, the model is trained to produce an output image in which **each object instance is assigned a distinct but uniform color**. For a diffusion model, we encode the image without adding noise, fix the timestep to $t = 999$ (the noisiest), and use the decoded image directly as the predicted instance segmentation (Garcia et al., 2024). For MAE, we similarly forward the image through the encoder and decoder without masking. This simple recipe enables broad zero-shot generalization with **one-step, deterministic, image segmenters**.

## 3.1 INSTANCE SEGMENTATION AS IMAGE TO IMAGE TRANSLATION

Recent instance segmentation models predict sets of binary masks, each representing an object instance (Carion et al., 2020; Cheng et al., 2022). However, it is not obvious how to enable generative models, designed to map from $\mathbb{R}^{W \times H \times 3} \to \mathbb{R}^{W \times H \times 3}$, to easily decode to this style. Drawing inspiration from work on image-to-image translation (Isola et al., 2017), we encode our ground truth as an RGB image with a unique color for each instance and black color for the background. We find that both the Stable Diffusion VAE and MAE models can decode these masks with effectively no loss in quality. Thus, we finetune our model by forwarding our RGB image (or its latent) without adding noise or masking, decoding the output, and optimizing in pixel space with respect to the RGB ground truth.

Unfortunately, one cannot simply assign each ground truth mask to a color and train the model to regress it since there are many possible accurate color assignments. Thus, we propose our instance coloring loss based on two key properties of an RGB segmentation mask. First, the color of all pixels within a mask should have low variance. Second, the color of a mask should not be predicted anywhere outside the mask. By emphasizing these two properties, we can learn a model of instance segmentation without needing to enforce specific colors for object masks. Simply, our finetuned model should ensure each instance is assigned a unique color that is consistent across its pixels. Our finetuning strategy draws on the same intuition as priors works that aim to cluster features for supervised instance segmentation (De Brabandere et al., 2017; Kong & Fowlkes, 2018). We designed it to enable a training strategy that is simple, intuitive, and agnostic to model architecture with fast inference.

More formally, assuming an image with $n$ instances, let $\Omega$ be the set of all pixels in the image, and $i \in \{0, \ldots, n\}$ the instance index where $i = 0$ is always the background. We define the set of pixels for instance $i$ as:

$$S_i = \{j \in \Omega \mid \text{pixel } j \text{ belongs to instance } i\}. \tag{1}$$

For each instance, we define the mean embedding (or representative color) as

$$\mu_{i,c} = \frac{1}{|S_i|} \sum_{j \in S_i} p_{j,c} \text{ and } \mu_{0,c} = 0; \forall i \in \{1 \ldots n\}, \forall c \in \{0, 1, 2\} \tag{2}$$

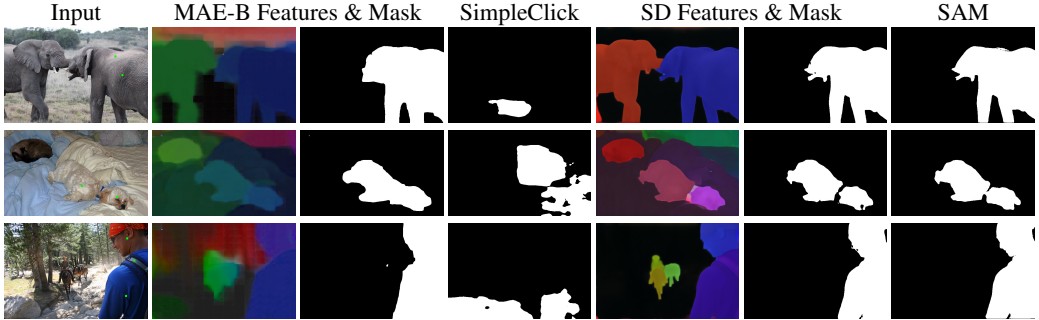

Figure 5: We showcase results for promptable segmentation using our features. Our finetuned MAE-B and SimpleClick are trained on the *same* data, using the *same* backbone, yet our MAE-B strongly outperforms SimpleClick due to its generative prior. Our finetuned Stable Diffusion has *never* seen a mask of the object type it is segmenting, but performs similar to SAM, which has been *heavily* supervised on over a billion masks of all types. *Prompt points are shown in green on the input.*

where $p_{j,c}$ is channel $c$ of the predicted color at pixel $j$. We force the background mask's mean to be black ($\mu_{0,c} = 0$) to follow standard convention and distinguish it from objects.

Our loss is composed of three components:

**1. Intra-Instance Variance Loss:** To ensure that the predictions within an instance are consistent, we use a smooth $\ell_1$ loss that encourages each pixel's prediction to be close to the instance mean. This is defined as

$$\mathcal{L}_{\text{var}} = \sum_{i=0}^{n} \frac{1}{|S_i|} \sum_{j \in S_i} \sum_{c=0}^{2} L_s(p_{j,c}, \mu_{i,c}) \tag{3}$$

where $L_s$ denotes the smooth $\ell_1$ loss. We find that using smooth $\ell_1$ loss over the standard $\ell_2$ loss converges better as it does not sharply penalize outliers.

**2. Inter-Instance Separation Loss:** We define the following loss to encourage the color of pixels outside the instance to be pushed away from the instance's mean color, ensuring that distinct regions do not converge to similar colors:

$$\mathcal{L}_{\text{sep}} = \sum_{i=0}^{n} \frac{1}{\sqrt{|S_i|}\,|T_i|} \sum_{j \in T_i} \frac{1}{1 + \sum_{c=0}^{2}(p_{j,c} - \mu_{i,c})^2} \tag{4}$$

where $T_i = \Omega \setminus S_i$ denote the set of pixels outside instance $i$. The loss is designed to saturate as the distance increases so that pixels far away from $\mu_i$ in color value do not dominate the loss. We include $\sqrt{|S_i|}$ in the denominator to emphasize smaller objects.

**3. Mean-Level Separation Loss:** To further separate instances, we design the following loss similar to the above one, but for mask centroids:

$$\mathcal{L}_{\text{mean}} = \frac{1}{n(n+1)} \sum_{0 \leq i < j \leq n} \frac{1}{1 + \sum_{c=0}^{2}(\mu_{i,c} - \mu_{j,c})^2} \tag{5}$$

where the first fraction simply normalizes by the total number of comparisons.

Finally, we finetune a pretrained diffusion model by minimizing our instance coloring loss $\mathcal{L}_{\text{IC}}$:

$$\mathcal{L}_{\text{IC}} = \mathcal{L}_{\text{var}} + \lambda_{\text{sep}}\mathcal{L}_{\text{sep}} + \lambda_{\text{mean}}\mathcal{L}_{\text{mean}} \tag{6}$$

where $\lambda_{\text{sep}}$ and $\lambda_{\text{mean}}$ are hyperparameters controlling the importance of each loss term.

### 3.2 POINT-PROMPTABLE INSTANCE SEGMENTATION

So far, our finetuned model assigns a color $F(x, y) = p_j$ to each pixel where $(x, y)$ is the location of pixel $j$. Since pixels belonging to the same object instance are encouraged to have similar colors,

| Model | COCO$_{\text{exc}}^{L}$ | COCO$_{\text{exc}}^{M}$ | COCO$_{\text{exc}}^{S}$ | DRAM | EgoHOS | iShape | PIDRay |
|---|---|---|---|---|---|---|---|
| SAM | 57.0 | 59.5 | 56.9 | 50.2 | 56.4 | 16.8 | 44.2 |
| SimpleClick | 1.4 | 0.6 | 0.2 | 2.4 | 1.6 | 1.6 | 1.5 |
| DINO-B | 35.0 | 11.0 | 1.7 | 29.4 | 14.8 | 27.4 | 14.9 |
| gen2seg (MAE-B) | 44.6 | 17.8 | 2.9 | 34.3 | 28.9 | 31.1 | 21.6 |
| gen2seg (MAE-H) | 50.0 | 23.2 | 3.5 | 40.3 | 31.9 | 34.9 | 24.1 |
| gen2seg (SD) | **57.6** | **38.8** | **8.5** | **48.2** | **40.0** | **51.4** | **30.9** |

Table 1: We evaluate zero-shot mIoU at a single prompt point on a wide spread of datasets. We match or recover a high percentage of performance (minimum 70%) on all datasets except COCO$_{\text{exc}}^{M}$/S. This suggests that, for larger objects, our models have learned strong object-level representations that transfer across categories and styles. Our baselines, SimpleClick and DINO-B are far below MAE-B, suggesting this generalization is unique to generative models. Additionally, we strongly outperform SAM on the iShape dataset, which evaluates segmentation of detailed and complex structures. **SAM is trained on SA-1B. All other models are trained on our limited Hypersim+VK2.**

we develop a simple method to "point-prompt" the feature map $F$ for binary masks (similar to SAM (Kirillov et al., 2023)). We intentionally opt not to train a separate mask decoder to showcase that our model's output features truly represent object instance shapes. Additionally, existing mask decoders use specialized architectures designed to upsample from low resolutions to the original image size, while our features are at the same resolution as the original image. Our simple prompting method can be viewed as analogous to similar evaluation methods in representation learning used to show that features truly represent the desired task, such as nearest-neighbor classification, semantic segmentation, or tracking (Caron et al., 2021). One can potentially improve our results by training a promptable high-resolution mask decoder on top of our features, which we leave for future work.

For each prompt point $p \in \mathbb{R}^2$ which contains the $x$ and $y$ location of the prompt pixel, we calculate the query vector $q_p \in \mathbb{R}^3$ as a Gaussian weighted average of the predicted color at the neighborhood of $p$ by $q_p = \frac{\sum_{x,y} w(x,y) F(x,y)}{\sum_{x,y} w(x,y)}$ where $w(x,y)$ is a Gaussian function with mean $p$ and standard deviation $0.01(W, H)$ and $W$ and $H$ are the width and height of $F$.

We then compute a query-feature similarity map:

$$S_p(x, y) = \min(1, \frac{1}{\|F(x,y) - q_p\|_2}),\tag{7}$$

normalize it between [0, 1], and smooth it with a joint bilateral filter (Petschnigg et al., 2004) (using $F$ as guidance), thus averaging the similarities close in both pixel location $(x, y)$ and feature value $F(x, y)$. Finally, assuming a set of $k$ point prompts, we take the per-pixel maximum across $k$ similarity maps, and threshold the merged similarity map to produce the binary mask.

## 4 EXPERIMENTS

### 4.1 DATASETS

**Training:** Inspired by previous work Ke et al. (2024), we train our model exclusively on synthetic data. We combine two datasets: Hypersim (Roberts et al., 2021) and Virtual Kitti 2 (Cabon et al., 2020). Hypersim provides a rich variety of indoor scenes (i.e. bathrooms, bedrooms, libraries, and kitchens), while Virtual Kitti 2 focuses on outdoor driving scenes, with annotations limited to cars. Both datasets are realistic and do not contain other styles. After removing images with too few masks, our dataset comprises of 86,000 images (66,000 from Hypersim and 20,000 from Virtual Kitti 2), with a sampling strategy that selects Hypersim images 90% of the time. Neither dataset includes annotations of people, animals, and several other categories. Additionally, while the number of images is comparable to existing instance segmentation datasets, the *diversity* is substantially lower. Our subset of Hypersim contains multiple views sampled from just 457 scenes, while Virtual Kitti 2 contains just 5 videos (each ∼15 seconds long), repeated from different viewing angles and weather conditions. A list of labeled object types in Hypersim is available in Appendix D.

| | Test Data | | | | | | |
|---|---|---|---|---|---|---|---|
| **Training Data** | $COCO_{exc}^{L}$ | $COCO_{exc}^{M}$ | $COCO_{exc}^{S}$ | DRAM | EgoHOS | iShape | PIDRay |
| Original | 50.0/57.6 | 23.2/38.8 | 3.5/8.5 | 40.3/48.2 | 31.9/40.0 | 34.9/51.4 | 24.1/30.9 |
| COCO | 53.6/64.0 | 18.8/44.0 | 2.9/9.8 | 48.1/51.2 | 26.8/35.2 | 33.4/41.2 | 25.7/31.9 |
| ClevrTex | 40.0/47.1 | 19.4/21.6 | 1.9/7.5 | 23.5/28.0 | 21.4/21.9 | 27.6/32.1 | 22.2/23.7 |
| 10 classes | 54.8/56.1 | 21.7/35.2 | 2.7/5.0 | 40.1/45.1 | 29.4/38.5 | 33.0/53.6 | 17.6/22.8 |
| 5 classes | 42.1/47.6 | 16.2/29.7 | 1.4/6.4 | 34.2/38.2 | 23.7/34.4 | 28.5/48.5 | 15.2/19.4 |

Table 2: We explore how the diversity of the finetuning domain impacts generalization. Interestingly, our model still performs well with real-world data (COCO) or synthetic shape-segmentation datasets (ClevrTex). Furthermore, performance with just 10 classes from Hypersim results in *nearly identical performance* to the full dataset (33+ classes), suggesting generalization emerges without diverse finetuning data. However, some degree of diversity is still needed for optimal performance, as seen by the performance drops on ClevrTex or with only 5 classes. We report results with both the gen2seg MAE-H and SD backbones as MAE-H/SD.

**Evaluation:** Our finetuned models have seen masks of some limited object categories in a single synthetic and realistic style. We aim to evaluate our models' zero-shot generalization to unseen categories and styles. We select 5 datasets from (Kirillov et al., 2023), each of which contains a very different domain from the others: **COCO_exc** (The COCO 2017 validation set (Lin et al., 2014), but we exclude object types seen in finetuning, The list of categories are presented in Appendix D), **DRAM** (Cohen et al., 2022) (art), **EgoHOS** (Zhang et al., 2022b) (egocentric), **iShape** (Yang et al., 2021) (complex and fine structures), and **PIDRay** (Zhang et al., 2022a) (luggage x-rays). *We describe the details and motivation for each dataset in Appendix A.2.*

## 4.2 Models and Baselines

Understanding the conditions under which generalization is possible is central to understanding our findings. In practice, generalization depends on two types of factors: the choice of model architecture and the properties of the training data. First, we implement our method on several generative models and compare them with some baselines. Unless specified, the models are finetuned using the loss and datasets described above. We finetune Stable Diffusion variants at $480 \times 640$ (Hypersim) and $368 \times 1024$ (Virtual Kitti 2) and ImageNet-pretrained models at $224 \times 224$.

We apply our method to four different settings that include diffusion and MAE models: **Stable Diffusion v2 (SD)** (Rombach et al., 2021): Following Garcia et al. (2024), we set $t = N-1$ in an $N$-step latent diffusion model and finetune end-to-end without noise. Using the frozen VAE to decode the U-Net output, we optimize pixel-space loss, yielding a one-step, deterministic image segmenter. **MAE with Decoder (MAE-B/H)** (He et al., 2022): We finetune an MAE with its decoder end-to-end to showcase that a strong generative prior learned solely from ImageNet-1K images without internet-scale pretraining or text supervision can effectively generalize. We use MAE ViT-B for direct comparison to DINO-B/SimpleClick (see below) and MAE ViT-H to demonstrate scalability.

We compare with the following baselines: **SimpleClick** (Liu et al., 2023a): A SOTA promptable segmenter using an MAE-B ViTDet (Li et al., 2022) backbone. We finetune SimpleClick using its released training code on our synthetic dataset to show that existing architectures cannot generalize well beyond supervised categories. **DINO + VAE (DINO-B)** (Caron et al., 2021): To test whether generalizable object groupings are exclusive to generative pretraining, we attach DINO to a frozen VAE decoder (from Stable Diffusion) via a simple up-conv and fine-tune end-to-end. DINO provides the discriminative features, while the VAE can decode to object shapes unseen in finetuning. **Segment Anything (SAM)** (Kirillov et al., 2023): We use SAM ViT-H off-the-shelf as a high-water mark for generalization that is supervised on the huge SA-1B dataset with 1B annotated masks. We examine the role of training data by varying both the data domain and number of object categories. We train on (i) the full MS-COCO train split or (ii) ClevrTex (Karazija et al., 2021), a synthetic, shape-centric dataset. COCO has real-world visual diversity but includes noisier labels while Clevr-Tex contains pixel-perfect annotations but a narrow set of simple object types. As our Hypersim and Virtual Kitti 2 mix contains both complex shapes and high quality labels, comparing these regimes

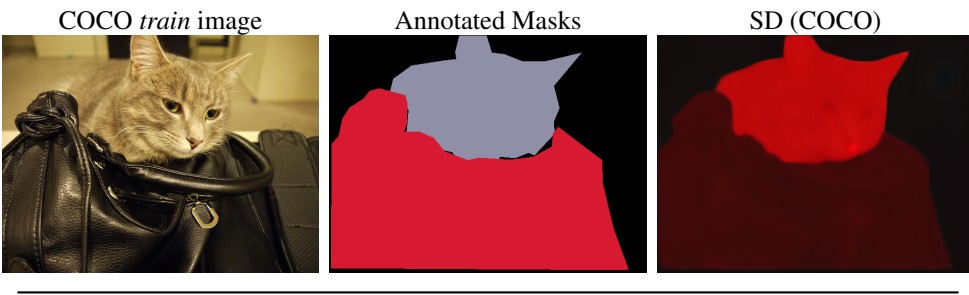

| | COCO *train* image | | Annotated Masks | | SD (COCO) | |

| Model | Edge AP | Model | Edge AP |
|---|---|---|---|
| DINO-B | 33.2 | gen2seg (MAE-H, ClevrTex) | 86.4 |
| gen2seg (MAE-B) | 53.7 | gen2seg (MAE-H, 10 classes) | 86.9 |
| Sobel | 65.0 | gen2seg (SD, ClevrTex) | 88.5 |
| gen2seg (MAE-H, COCO) | 75.9 | gen2seg (SD, COCO) | 89.7 |
| SAM | 79.0 | gen2seg (SD, 5 classes) | 91.7 |
| gen2seg (MAE-H) | 80.5 | gen2seg (SD, 10 classes) | 92.5 |
| gen2seg (MAE-H, 5 classes) | 83.1 | gen2seg (SD) | 93.4 |

Figure 6: We evaluate the zero-shot edge AP for recall less than 20% of the edges synthesized from our models and baseline segmentations on BSDS500. Nearly all generative models outperform SAM, even when trained on polygonal mask edges from COCO. *Observe how in the figure above, even though the annotation has very noisy edges, SD (COCO) does not learn this and instead predicts smoother and more perceptually aligned boundaries.* This suggests that our models' ability to produce detailed groupings is due to the generative prior, and not any specific dataset.

helps disentangle the impact of object complexity from annotation quality. To study category diversity, we also experiment with restricting Hypersim to only 10 and 5 classes of labeled masks in finetuning. To ensure the model does not see the masks of unknown categories, we disable the loss for pixels within the bounding box of all unknown objects. *Additional details regarding models and datasets appear in Appendix A.1.*

## 4.3 ZERO-SHOT PROMPTABLE INSTANCE SEGMENTATION

We evaluate our model on the task of zero-shot point-promptable instance segmentation. Following Kirillov et al. (2023), we first evaluate our models' ability to produce reasonable masks using only a single prompt point at the ground truth object center and comparing the predicted mask's IoU with the respective ground truth. Then, following the so-called "golden" standard of prompting (Ravi et al., 2024; Kirillov et al., 2023; Liu et al., 2023a; Sofiiuk et al., 2022; Lin et al., 2022; Sofiiuk et al., 2020), we iteratively find the largest contiguous area of the ground truth with no mask predicted yet, and select the next prompt point in that area closest to the area's center. As promptable segmentation is a highly ambiguous task, we do not necessarily expect to get high IoU, but we do hope for it to approach our high-water mark, SAM.

**Results.** First, we examine model performance using a single prompt at the object center (Table 1). On all evaluation datasets except $COCO_{exc}^{M/S}$ (Medium and Small), our model approaches or marginally exceeds SAM, despite never seeing labeled masks for these categories. This indicates that generative models learn transferable object-level features, particularly for larger objects and intricate details (evidenced by superior results on the iShape dataset). Interestingly, SAM struggles in some cases because it seems to group by texture for out-of-distribution cases, such as art.

Our method also surpasses both baselines, SimpleClick and DINO-B. SimpleClick, as expected, struggles with zero-shot mask generation. SimpleClick's failure to generalize represents a weakness in existing segmentation architectures: nearly all models use mask predictors finetuned from scratch (Kirillov et al., 2023; Wang et al., 2024). When faced with object types unlike anything seen in finetuning, these models will fail to generalize as the mask predictor has only seen the object types in finetuning and lacks any other visual priors. Our method deliberately uses only generatively pretrained parameters, so that the *entire model* retains broad visual priors for test-time generalization. Our finetuned DINO-B model successfully activates on objects, but struggles to separate their instances. We hypothesize this is because self-distillation (and discriminative pretraining in general)

over-emphasize semantics via *invariant* representations, meaning they enforce that the output representation does not change across augmentations. In contrast, to succeed at instance segmentation, one must learn *equivariant* representations, meaning they account for changes in the scale, shape, or structure of the image (and the objects within) (He, 2017). We hypothesize that generative models are well posed to learn equivariant representations because they must learn to synthesize a plausible image for every corrupted input they receive.

However, our models have limitations segmenting small objects, likely due to biases from pretraining: Stable Diffusion's text conditioning emphasizes large, prominent objects, while MAE's ImageNet-1K pretraining prefers central "main" objects. Additionally, models like SAM and other state-of-the-art segmenters (He et al., 2017; Cheng et al., 2022; Cai & Vasconcelos, 2019) fine-tune at $1024 \times 1024$ resolution, whereas we fine-tune at lower resolutions: $480 \times 640$ (Hypersim) and $368 \times 1024$ (Virtual Kitti 2) for Stable Diffusion variants, and $224 \times 224$ for ImageNet-based models. We expect that stronger generative models, such as FLUX.1, would enhance small-object segmentation. We leave this to future work due to their large parameter counts.

We also find that our models generalize well, even without synthetic or complex datasets, as shown in Table 2. Surprisingly, our models learn to generalize *even with only 5 object types (books, chairs, lamps, tables, and pillows) seen in finetuning,* or when only finetuned on simple shapes such as cubes or spheres (ClevrTex). Furthermore, our models still excel at segmenting fine structures (as shown on iShape) when finetuned on COCO, which mostly contains polygonal mask annotations. Additionally, we see only minor improvement when generative models are finetuned with COCO, suggesting that our models' zero-shot generalization is very close to the "upper bound" of seeing the objects in finetuning. This suggests generative pretraining is a powerful paradigm to learn generalizable instance grouping priors.

### 4.4 ZERO-SHOT EDGE DETECTION

We evaluate our segmentation features on the task of edge detection with the BSDS500 dataset (Martin et al., 2001). It is important to note that we are not simply trying to find all edges in the image, but only *object boundaries*. Following Kirillov et al. (2023), we simply apply a Sobel filter on the predicted features followed by nonmax suppression to thin the edges. We use the edge detection described above on the original image as a "weak" baseline and on the output of SAM's AutoMaskGenerator as a "strong" baseline. We report AP for recall less than 20%. We explain this choice (and display the full precision-recall curves) in Appendix B.

**Results.** Our model accurately delineates the edges of primary objects in each image despite never seeing a mask of the object type. As shown in Table 6, our SD model produces much finer edges compared to SAM. Additionally, our MAE-H model marginally outperforms SAM despite being supervised at less than 5% of the resolution ($224 \times 224$ vs $1024 \times 1024$). We also observe that when we change the training data from synthetic Hypersim+VK2 to human-labeled COCO, we observe a decrease of less than 5 points in edge AP on both SD and MAE-H models. Because COCO's edges are coarse and polygonal, the relatively small performance drop supports our hypothesis that our model's accurate and clean edges stem from generative pre-training and are not due to the bias from the synthetic data. More interestingly, our SD (COCO) model is still better than SAM by more than 10 points for recall less than 20%. Both COCO and SA-1B have similar polygonal edges, which suggests by learning to synthesize scenes, generative models inherently learn a detailed representation of object boundaries. Our precise edges also persist (and are sometimes even stronger than the base model) when dataset complexity reduces, suggesting the generative model "defaults" to predicting clean edges when segmenting an object.

## 5 CONCLUSION

Our findings suggest that learning to generate visual reality inherently teaches a detailed understanding of its constituent parts. Our models are able to segment objects and styles *nothing like* anything any labels seen in finetuning, yet still achieve competitive performance with heavily supervised models. As we continue to scale generative models and diversify their pretraining data, their ability to perceive the visual world will only grow. Learning to leverage these powerful representations for a wide variety of visual tasks has the potential to enable a new frontier in generalizable perception.

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

# APPENDIX

## A  EXPERIMENTAL MODELS AND DATASETS

### A.1  ADDITIONAL DETAILS FOR MODELS AND BASELINES

**Stable Diffusion 2 (SD)**: We finetune Stable Diffusion 2 U-Net end-to-end using our loss. We follow the method in (Garcia et al., 2024) which enables deterministic one-step prediction. This has been shown to outperform multi-step stochastic inference with standard diffusion training for other perceptual tasks. To train it in pixel space, we fix the timestep to the highest (999). We replace the input with our image's VAE latent without adding any noise. We set the CLIP embedding to the null condition. We freeze the VAE and finetune the U-net end to end with our instance coloring loss as described above. We show in Appendix B that the choice of timestep has negligible effect on the results as the timestep cross-attention within the U-net is finetuned too.

**MAE with Decoder (MAE-B/H)**: We finetune MAE with the decoder end-to-end. In particular, MAE is trained *only* on images, so it allows us to explore if a strong generative prior (without any additional labels such as text or class condition) is enough to learn strong segmentation features. Additionally, we can evaluate if internet-scale pretraining is necessary to generalize. We finetune both the ImageNet-1K pretrained MAE ViT-B (for a direct comparison with DINO and SimpleClick) and the ViT-H (for a *rough* comparison to SAM and Stable Diffusion). The masking ratio is set to 0%, so no tokens are masked. Our backbones use normalized tokens during pretraining, as this is the standard. We hypothesize this leads to the token artifacts seen in the qualitative figures. It is possible results may improve by using un-normalized tokens.

**SimpleClick**: We train SimpleClick, a state-of-the-art promptable instance segmenter, on the same synthetic datasets that we train our generative models on. This helps us investigate whether our generalization is due to the generative model itself, or simply an effect of training on synthetic data. Our model initializes with an MAE ViT-B encoder, along with a feature pyramid learned from scratch. We finetune using its released training code.

**DINO + VAE (DINO-B)**: We investigate whether a strong discriminative model can suffice for segmentation features without necessarily learning to synthesize images. However, we must pair it with a model that knows how to upsample images of all types from a low-dimensional space so that it will be able to generate images of objects unseen in training. To investigate this, we join ImageNet-1K pretrained DINO ViT-B with the Stable Diffusion VAE decoder, and connect them with a single up-conv layer. Additionally, the latent features of images compressed with the Stable Diffusion VAE often look like the image itself (Kouzelis et al., 2025). Thus, to succeed, all DINO needs to do is generate these object groupings at a lower resolution.

**Segment Anything (SAM)**: SAM is a large-scale promptable instance segmentation model supervised on over a billion masks from a very large distribution of data. We use SAM as a benchmark to evaluate the extent to which our model generalizes across a wide variety of images. While in some cases we outperform SAM, we do not mean for this to claim that our models are inherently superior to it. We use the publicly available ViT-H checkpoint and inference library.

**Dataset Baselines**: We finetune the models (MAE-H and SD) with the same loss and implementation details. For 5 and 10 category models, we computed no loss for any pixels inside the bounding boxes of the objects that are outside the "allowed" categories. We also pruned any images with more than 35% of image area that is "not allowed" or any images with no allowed masks as we found it important to have a smaller amount of high quality data, rather than a larger set of low quality data.

To match their pretraining resolutions, SimpleClick, DINO, and MAE are finetuned at $224 \times 224$ resolution, rather than the standard ones described in implementation details.

### A.2  EVALUATION DATASETS

**COCO$_{exc}$**: We evaluate on the COCO 2017 validation set, as is standard for a large variety of segmentation tasks. To showcase our zero-shot generalization to unseen mask categories, we choose a subset of COCO dataset with categories not seen in our finetuning and call it COCO$_{exc}$ dataset. COCO$_{exc}$ does not include object types that Hypersim does not have a explicit category for, but we

have observed to exist in the dataset (i.e. wine glass, teddy bear, potted plant). However, we have seen that our synthetic data does not contain any masks for humans or animals, so COCO$_{\text{exc}}$ includes such images. COCO$_{\text{exc}}$ contains 86% of the images and 64% of the masks in the full COCO 2017 validation set. We provide a full list of categories in Appendix D.

**DRAM**: Humans are able to perceive objects in a wide variety of visual media with large amounts of abstraction. Unfortunately, most existing segmentation methods require high amounts of labeled data to generalize to art. DRAM is a segmentation dataset that has annotated a large variety of art pieces across styles and time periods, including many abstract styles such as impressionism, ink-and-wash, and cubism. We evaluate on the test set.

**EgoHOS**: Egocentric vision is crucial for embodied AI and robotics. However, segmentation in egocentric tasks is challenging because the first-person views often have frequent occlusions, motion blur, and variable lighting, resulting in inconsistent and ambiguous object boundaries. The EgoHOS dataset provides segmentations of a large amount of egocentric images of humans interacting with everyday objects. We evaluate on the test set.

**iShape**: Many objects in the real world have fine, intricate structures. We evaluate our models' ability to accurately segment these objects using the iShape dataset. iShape is an instance segmentation dataset which contains many occlusions and complex, fine structures such as wires or fences. We evaluate on the test set using the PNG Cityscapes style annotations.

**PIDRay**: Humans effortlessly apply their understanding of object shapes even in scenarios never encountered in nature. For instance, TSA agents can quickly identify dangerous items in X-ray images of luggage. To assess our models' generalization to this challenging scenario, we employ the PIDRay dataset, which features labeled examples of hazardous objects in baggage. This task is particularly demanding because many dangerous items are small and deliberately concealed within other objects. We evaluate on the "easy" subset of the test set.

# B  ADDITIONAL RESULTS AND EXPERIMENTAL DETAILS

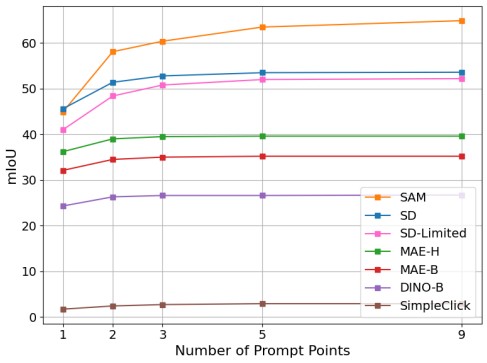

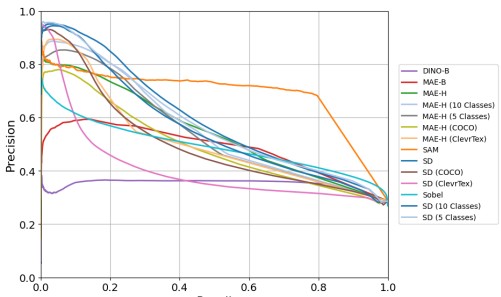

Figure 7: We evaluate segmentation quality as the number of prompt points increases. Our SD model marginally exceeds SAM at 1 prompt point and recovers >82% of SAM's performance at 9 prompt points. This is surprising as we do not use a learned mask encoder for multiple prompt points, but simply merge similarity maps computed individually from each point prompt.

Figure 8: We plot the full precision-recall of the zero-shot edge detection on BSDS500. Our strongest models outperform SAM's precision when recall is low, suggesting their segmentations lie on the exact boundaries of the corresponding objects. However, as shown in Figure 9, our models sometimes do not predict objects for certain regions of the image, leading to lower precision at higher recall values.

We explore if our masks improve with additional prompt points (Figure 7). We average IoU across datasets, excluding COCO$_{\text{exc}}^{M/S}$ (since few small objects are detected in the first place). Remarkably, our SD model achieves over 82% mIoU relative to SAM at 9 prompt points, despite lacking a learned prompt encoder for multi-point prompts or masks from any of the evaluated categories.

| Original | GT | SD | Edges | MAE-H | Edges | SAM | Edges |
|---|---|---|---|---|---|---|---|

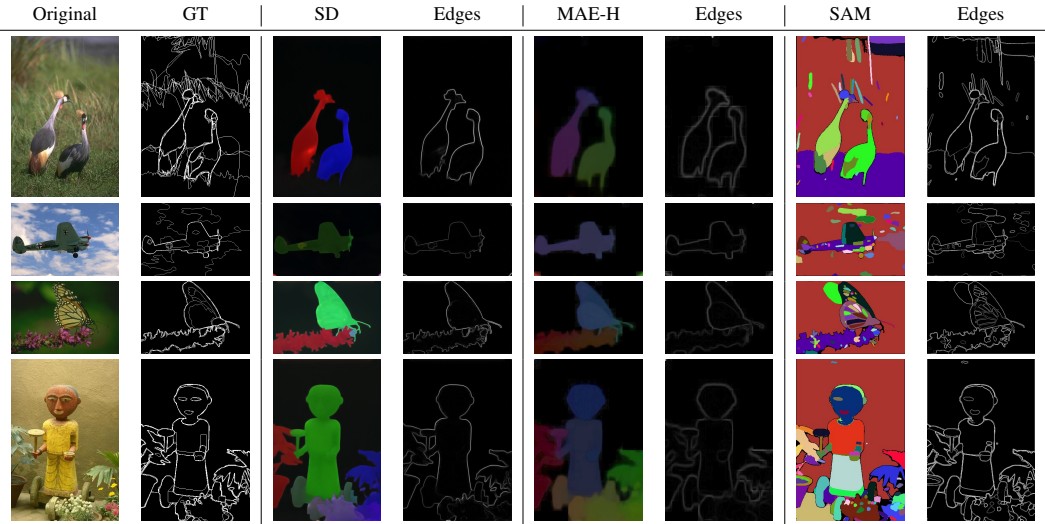

Figure 9: Delineating the edges of the *objects* in the scene is a fundamentally ambiguous task. While our models' outputs do not exactly match the ground truth (neither do SAM's), they represent one interpretation of the "objects" in the scene. Our model tends to include certain objects, such as the clouds or grass, in the background. This emerges without supervision and may be an inherent bias from generative pretraining.

| $\mathcal{L}_{\text{var}}$ | $\mathcal{L}_{\text{sep}}$ | $\mathcal{L}_{\text{mean}}$ | smooth $\ell_1$ | Norm | mIoU |
|---|---|---|---|---|---|
| ✗ | ✓ | ✓ | ✓ | ✓ | 0.2 |
| ✓ | ✗ | ✓ | ✓ | ✓ | 30.4 |
| ✓ | ✓ | ✗ | ✓ | ✓ | 29.3 |
| ✓ | ✓ | ✓ | ✗ | ✓ | 27.7 |
| ✓ | ✓ | ✓ | ✓ | ✗ | 26.2 |
| ✓ | ✓ | ✓ | ✓ | ✓ | **31.6** |

Table 3: Ablation study using mIoU at a single center point on COCO$_{\text{exc}}$ on our SD model.

| SD timestep | COCO$^L_{\text{exc}}$ | COCO$^M_{\text{exc}}$ | COCO$^S_{\text{exc}}$ | DRAM | EgoHOS | iShape | PIDRay |
|---|---|---|---|---|---|---|---|
| t=999 (original) | 57.6 | 38.8 | 8.5 | 48.2 | 40.0 | 51.4 | 30.9 |
| t=499 | 58.8 | 37.5 | 6.2 | 48.9 | 28.6 | 50.9 | 31.5 |
| t=1 | 57.9 | 38.4 | 8.5 | 47.8 | 31.3 | 49.2 | 28.0 |

Table 4: We ablate the role of the fixed timestep in our results. As shown above, it does not make a major difference on the results as the timestep cross-attention is finetuned in the U-net.

Additionally, as shown in Figure 8, our Stable Diffusion variant outperforms SAM in the first quarter of the precision-recall curve, and our MAE ViT-H variant matches SAM for the first 20% of recall despite having a substantially smaller training resolution when evaluated on edge accuracy on BSDS500. However, as visualized in Figure 9, many labeled regions exist at the interface between object and background (e.g., clouds, plants, or rocks). Our model tends to include these regions in the background. As a result, no edges are detected for some objects in our features, and the precision falls for higher recalls. This behavior is fully emergent, as our model has never seen masks of the overwhelming majority of objects present in the BSDS500 dataset. This hypothesis is further supported by how our models with 5 and 10 classes only perform better. Specifically, they have no loss computed on a high amount of the background (because we ignore loss for pixels inside the bounding box of unknown-category objects). As a result, our 5 and 10 category models sometimes activate on objects that SD or MAE-H would consider background. Thus, we opt to report AP under

| Augmentation | gen2seg (MAE-H) | | | gen2seg (MAE-H) | | |
|---|---|---|---|---|---|---|
| | $\text{COCO}_{\text{exc}}^{L}$ | $\text{COCO}_{\text{exc}}^{M}$ | $\text{COCO}_{\text{exc}}^{S}$ | $\text{COCO}_{\text{exc}}^{L}$ | $\text{COCO}_{\text{exc}}^{M}$ | $\text{COCO}_{\text{exc}}^{S}$ |
| Original | 57.6 | 38.8 | 8.5 | 50.0 | 23.2 | 3.5 |
| Solarize | 58.6 | 35.5 | 7.7 | 50.8 | 21.3 | 1.1 |
| Contrast 2.0× | 53.6 | 36.7 | 8.6 | 46.7 | 19.7 | 2.4 |
| Contrast 0.5× | 61.4 | 38.4 | 8.5 | 49.6 | 27.6 | 3.2 |
| Grayscale | 56.7 | 34.0 | 7.7 | 45.5 | 23.0 | 3.1 |
| Hue +0.3 | 55.8 | 34.2 | 7.9 | 49.7 | 21.8 | 1.4 |
| Hue -0.3 | 57.0 | 35.9 | 8.2 | 51.0 | 23.2 | 3.5 |

Table 5: We evaluate our models' zero-shot accuracy robustness to several image color changes and perturbations. As shown below, our models exhibit strong robustness to all with limited drop in mask quality. Particularly, the solarize augmentation results loosely suggest that our model is not grouping based on low-level color/texture and has some sense of high-level grouping.

| Last N blocks finetuned | $\text{COCO}_{\text{exc}}^{L}$ | $\text{COCO}_{\text{exc}}^{M}$ | $\text{COCO}_{\text{exc}}^{S}$ |
|---|---|---|---|
| MAE-H (original, full finetune) | 50.0 | 23.2 | 3.5 |
| 8 layers (decoder only) | 53.5 | 20.2 | 3.0 |
| 4 layers | 45.5 | 16.3 | 2.5 |
| 2 layers | 40.0 | 13.0 | 2.0 |
| 1 layer | 36.4 | 10.6 | 1.4 |
| 0 layers (linear probe) | 19.6 | 2.7 | 0.0 |

Table 6: We explore how many layers need to be finetuned for our method to work. Surprisingly, full finetuning does not yield performance gains over finetuning the decoder. Additionally, just one or two layers need to be finetuned for reasonable performance.

20% recall as the lower precision later in the curve is not due to inaccurate edges, but rather a lack of segmented objects in those regions.

We present an ablation study in Table 3 to examine the effect of each loss component and the VAE normalization using mIoU at a single center point on the COCO$_{\text{exc}}$ dataset. First, removing the intra-instance variance loss ($\mathcal{L}_{\text{var}}$) (row 1) causes performance to collapse. This is essential to our method; without it, the model does not produce uniform masks. Eliminating the pixel-level separation loss (row 2) prevents the model from learning sharp object boundaries, causing it to slightly overestimate mask boundaries. Eliminating the mean-level loss (row 3) results in a reduced mIoU as well, but instead affects the model's ability to discriminate smaller objects. Replacing the smooth $\ell_1$ loss with $\ell_2$ loss (row 4) and the normalization of VAE outputs (Norm, row 5) results in mIoU of 27.7 and 26.2, respectively. Both smooth $\ell_1$ loss and normalization of VAE output helps the model converge to lower loss values earlier. Overall, the complete model that integrates all these components (row 6) achieves the highest mIoU of 31.6, confirming that each element plays a role in obtaining optimal performance. We also ablate the fixed timestep value for our SD model in Table 4.

In Table 5, we explore our models' robustness to a variety of image perturbations. This helps show that our model is robust to changes in color and texture, despite never receiving any supervision for this.

We also explore how deep we need to finetune our model to achieve strong results in Table 6. We start with full finetuning and gradually reduce the layers trained to the last N layers in the model, plus the final projection to pixel space, freezing the rest of the model. We find that finetuning just the decoder leads to very similar performance to full finetuning. Additionally, we find that just one or two layers are enough to achieve reasonable performance. This suggests that the "core" object grouping information is highly saturated in the decoder, especially the last layers. This makes sense because 1) since the rest of the model is frozen, only the representations that are available at the last N layers will actually have an effect on the output 2) these layers are the ones most responsible for synthesizing images, for which understanding object grouping is frozen.

| Class | No Compositionality | gen2seg (MAE-H) | gen2seg (SD) | SAM |
|---|---|---|---|---|
| aeroplane (19) | 17.5 | 10.3 | 19.7 | 48.6 |
| bird (13) | 15.6 | 14.2 | 20.3 | 36.3 |
| bicycle (6) | 19.5 | **28.4** | **27.6** | 46.6 |
| bus (46) | 12.7 | 7.5 | 9.5 | 56.0 |
| cat (17) | 10.7 | **22.5** | **29.0** | 39.1 |
| cow (19) | 11.4 | 11.0 | **17.3** | 26.9 |
| dog (18) | 11.0 | **19.0** | **25.9** | 34.2 |
| horse (21) | 9.2 | 10.7 | **17.8** | 26.2 |
| motorbike (7) | 15.5 | **21.1** | **29.1** | 54.0 |
| person (24) | 9.2 | **27.3** | **31.0** | 54.1 |
| sheep (19) | 14.4 | 10.0 | 15.5 | 27.1 |
| train (52) | 30.9 | 25.0 | 34.0 | 38.7 |
| bottle (2) | 53.4 | 28.0 | 34.4 | 53.1 |
| car (30) | 18.7 | 10.6 | 13.6 | 57.7 |
| potted plant (2) | 54.4 | 36.2 | 38.4 | 58.4 |
| TV monitor (1) | 55.5 | 51.3 | 57.4 | 66.9 |

Table 7: **Emergent part-based grouping on Pascal-Part.** We evaluate part-level grouping on the Pascal-Part dataset (part masks for PASCAL VOC 2012) by measuring IoU between discovered part groups and ground-truth part masks. "No Compositionality" is a baseline computed as the IoU between each part mask and its corresponding whole-object mask (e.g., cat head vs. whole cat), approximating performance with no part-level grouping. Bold indicates classes where MAE-H or SD improves by at least 5 IoU points over the baseline. Gray rows denote classes included in our Hypersim+VK2 training mix.

Finally, in Table 7, we quantitatively evaluate emergent part-based grouping on the Pascal-Part Chen et al. (2014) dataset, which annotates part-level masks for the PASCAL VOC 2012 set. Bolded values represent classes for MAE-H or SD where our model performed at least 5 points better than the baseline.

We compare against a baseline "No Compositionality", which is calculated by evaluating IoU between every part mask and its whole object mask. This represents the values close to what we would get if there was no part-level grouping at all (i.e. predicting the whole cat when ground-truth is just the cat head). We also report results from SAM to serve as a high-water mark and provide context for our results.

We see signs of part-level grouping for some classes such as bike, cat, cow, dog, horse, motorcycle, and person. This emerges without any supervision on part or whole-object masks of these categories, suggesting it is inherent to the generative model.

Classes colored gray are included in our Hypersim+VK2 mix; thus we do not expect to see any improvement on them as the whole-object supervision may overwrite knowledge of part-level groups.

## C  Implementation details

We train our models on a node with four RTX6000 Ada GPUs. We train our models with a batch size of 2, with gradient accumulation for 4 steps, for an effective batch size of 32. We do this intentionally, as described in (Ke et al., 2024), to mix gradients between images sampled from Hypersim and Virtual Kitti 2. We train our model for 30,000 iterations, which takes about 29 hours for SDv2 and 12 hours for MAE ViT-H. However, our models show no signs of overfitting, and performance would likely benefit from additional iterations, but we didn't explore this due to timing constraints. We sometimes struggle with memory constraints when finetuning Stable Diffusion, as some images in Hypersim have a very high number of instances (2000+). Thus, we compute the loss on up to 1250 instances at most. We normalize final pixel-level outputs as we observe it improves convergence. We start our learning rate at 6e-5, after 100 steps of warmup. We then decay on a cosine schedule so that we end at $\frac{1}{20}$ of our original learning rate. We train at a resolution of

480×640 for Hypersim and 368×1024 for Virtual Kitti 2 for our Stable Diffusion variants. For ImageNet-pretrained models (MAE, DINO, SimpleClick), we resize Hypersim images to 224×224 and randomly crop a 224×224 region in Virtual Kitti 2 dataset. We set $\lambda_{\text{sep}}$ and $\lambda_{\text{mean}}$ to 300 and compute all losses in the range of $[0, 255]$ to weight all terms equally. For Stable Diffusion, we finetune only the U-net and freeze the VAE. We set the text condition to the empty string. For all prompting experiments, we fix the threshold to $\frac{3}{255}$ and use a window size of 9 (for joint bilateral smoothing). We also performed some minor data cleaning prior to training where we removed all images with no masks and any scenes with less than 10 objects.

# D LIST OF OBJECT TYPES IN HYPERSIM AND COCO$_{\text{EXC}}$

For each dataset, we provide a list of labeled object types present, along with the number of objects with that type.

## D.1 HYPERSIM

This list includes only objects with instance *and* class annotations. Certain objects which have instance-level annotations but lack class labels, such as teddy bears or potted plants, were placed into the "unknown" object category.

**Total number of objects:** 3,693,970

**Objects by category:** Unknown (1,375,739), books (1,149,313), chair (334,422), lamp (211,409), table (102,093), pillow (74,230), window (51,444), picture (46,102), cabinet (38,253), paper (34,420), sofa (30,895), blinds (29,462), clothes (28,917), door (20,410), box (19,769), desk (19,418), floormat (18,879), counter (14,446), bookshelf (12,887), shelves (11,886), sink (10,026), mirror (7,488), bed (6,001), towel (5,250), television (4,367), nightstand (2,803), bathtub (2,556), refrigerator (2,243), curtain (2,066), toilet (1,068), dresser (717), wall (106), whiteboard (100).

## D.2 COCO$_{\text{EXC}}$

We excluded object categories that either labeled in our train set or we have observed to appear. For example, we have excluded "wine glass" or "knife" as these would be placed into Hypersim's "unknown" category. We have personally verified all of the objects below to not exist in the subset of Hypersim we train on.

**Total number of objects:** 23,195

**Objects by category:** Person (11,004), traffic light (637), handbag (540), bird (440), boat (430), truck (415), umbrella (413), cow (380), banana (379), motorcycle (371), backpack (371), carrot (371), sheep (361), donut (338), kite (336), bicycle (316), broccoli (316), cake (316), suitcase (303), orange (287), bus (285), pizza (285), horse (273), surfboard (269), zebra (268), sports ball (263), elephant (255), tie (254), skis (241), giraffe (232), tennis racket (225), dog (218), cat (202), train (190), skateboard (179), sandwich (177), baseball glove (148), baseball bat (146), airplane (143), hot dog (127), frisbee (115), fire hydrant (101), stop sign (75), bear (71), snowboard (69), parking meter (60).

# E    ADDITIONAL QUALITATIVE SAMPLES

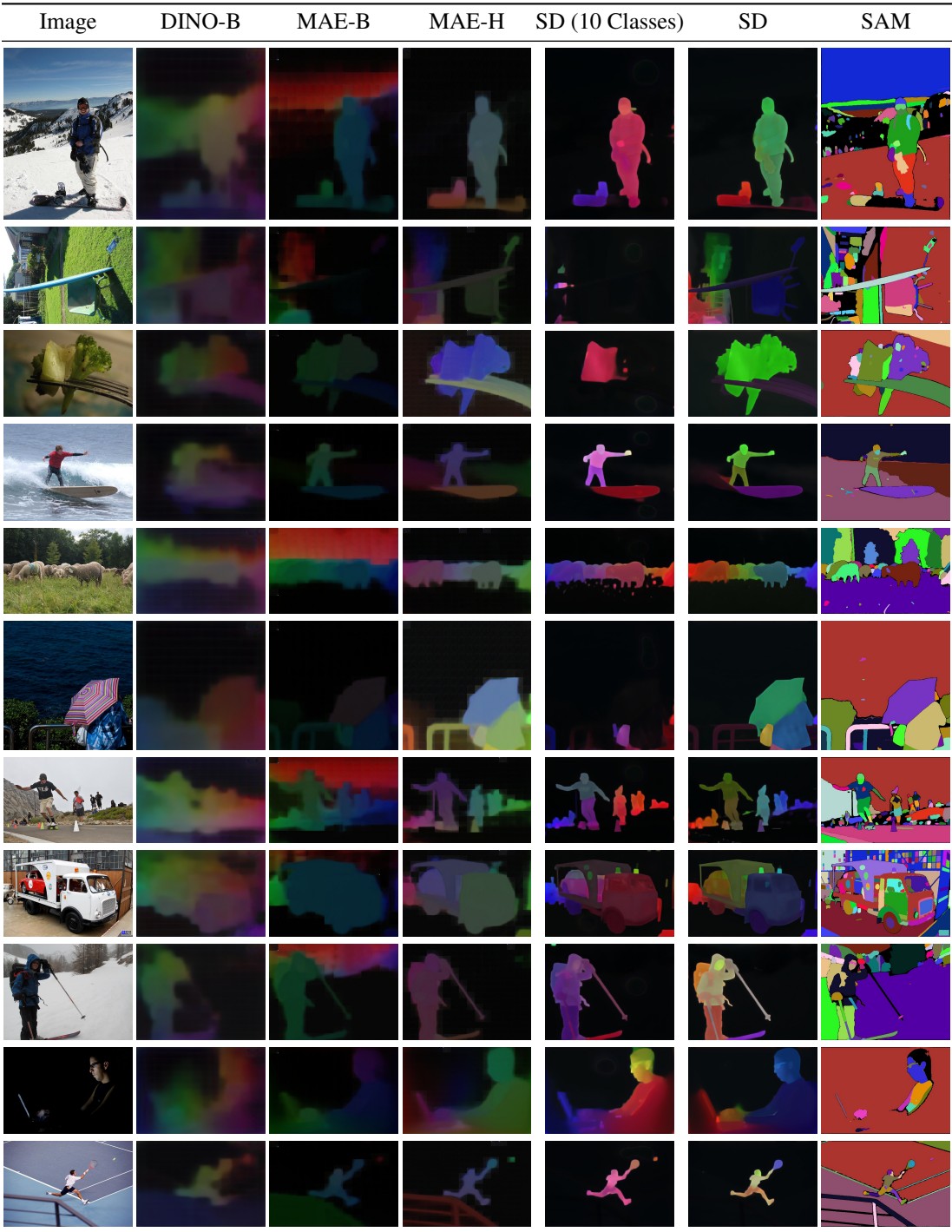

Figure 10: Qualitative Results on the COCO$_{exc}$ (Lin et al., 2014) dataset. These results are randomly chosen and not cherry-picked.

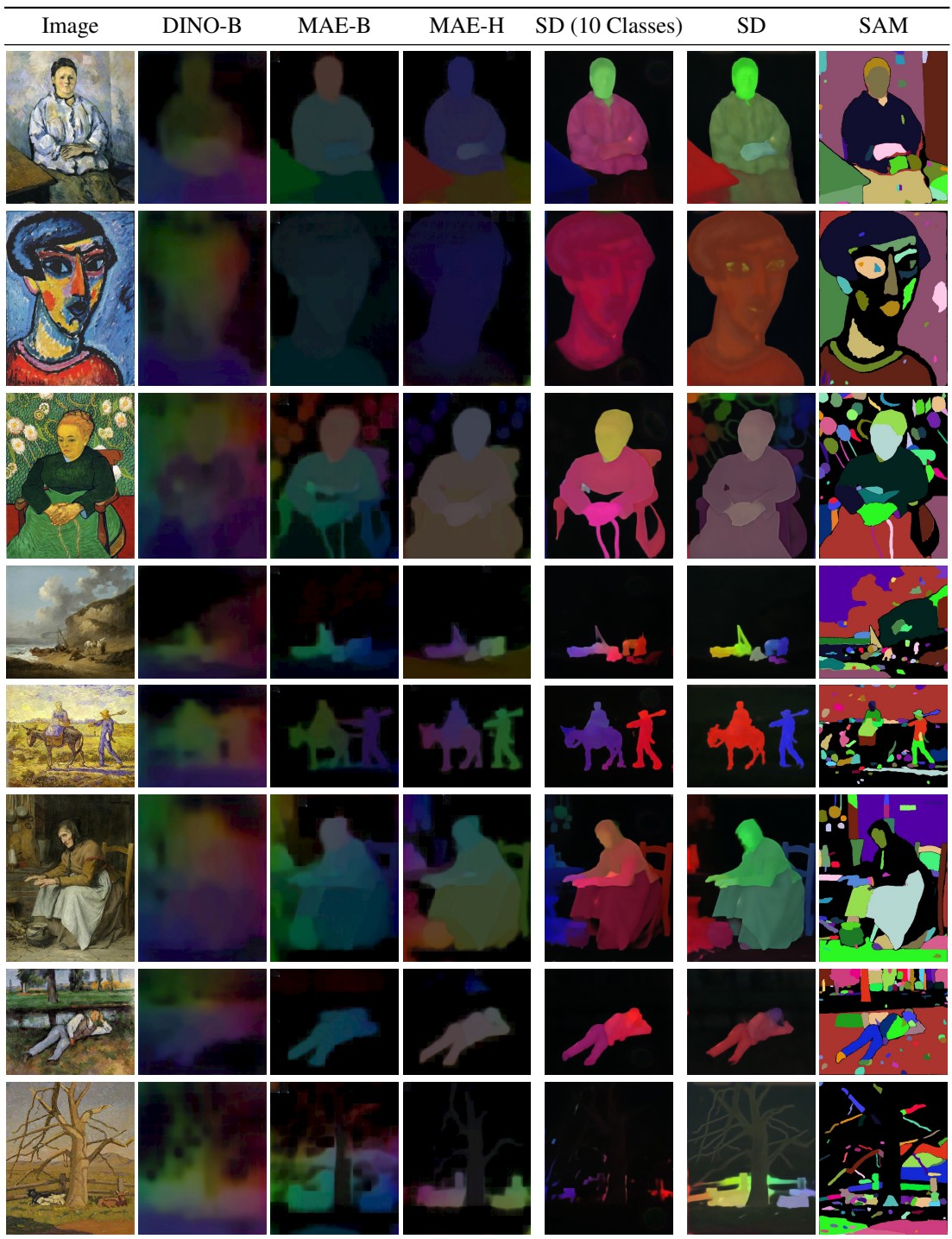

Figure 11: Qualitative Results on the DRAM (Cohen et al., 2022) dataset. These results are randomly chosen and not cherry-picked.

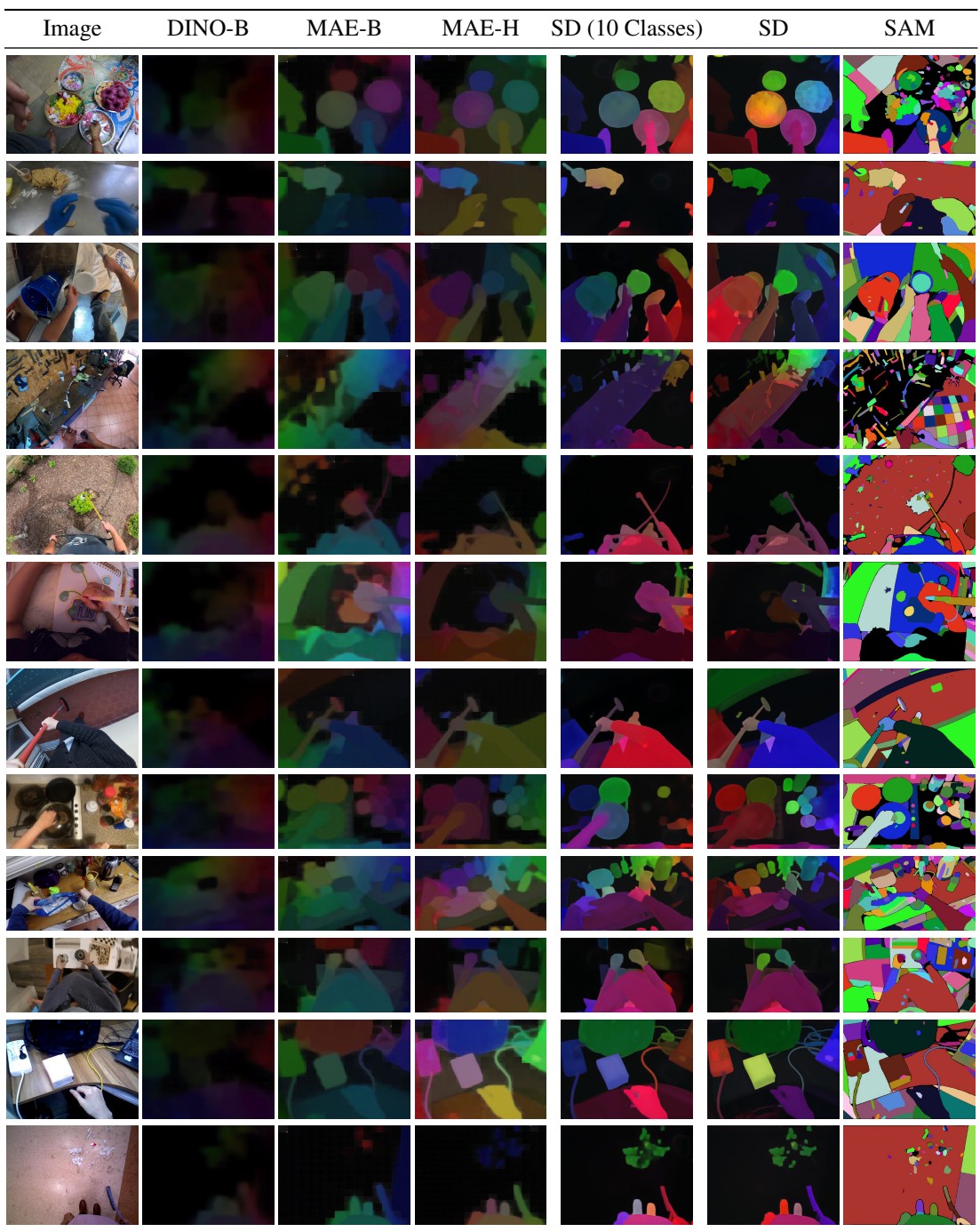

Figure 12: Qualitative Results on the EgoHOS (Zhang et al., 2022b) dataset. These results are randomly chosen and not cherry-picked.

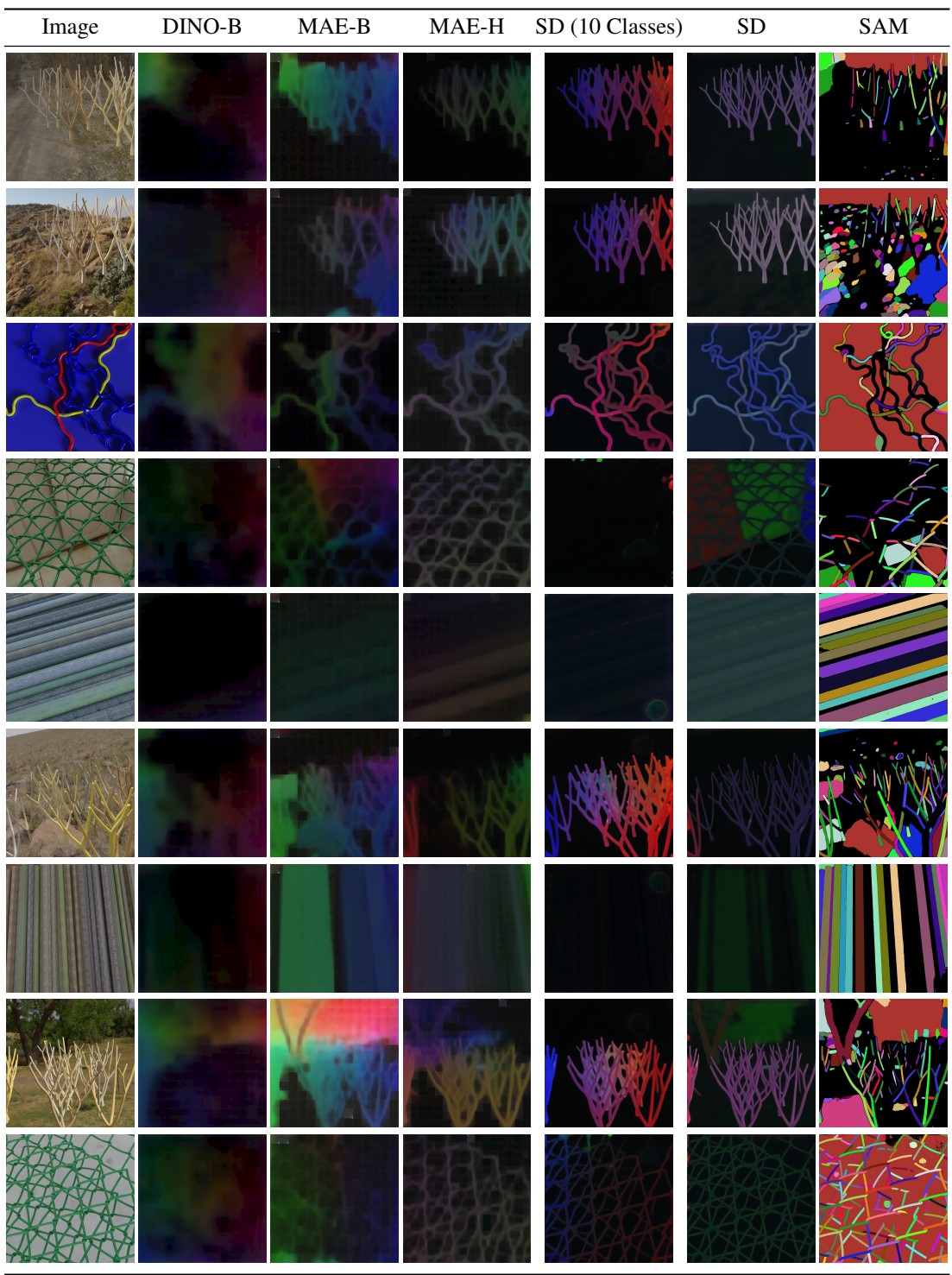

Figure 13: Qualitative Results on the iShape (Yang et al., 2021) dataset. These results are randomly chosen and not cherry-picked.

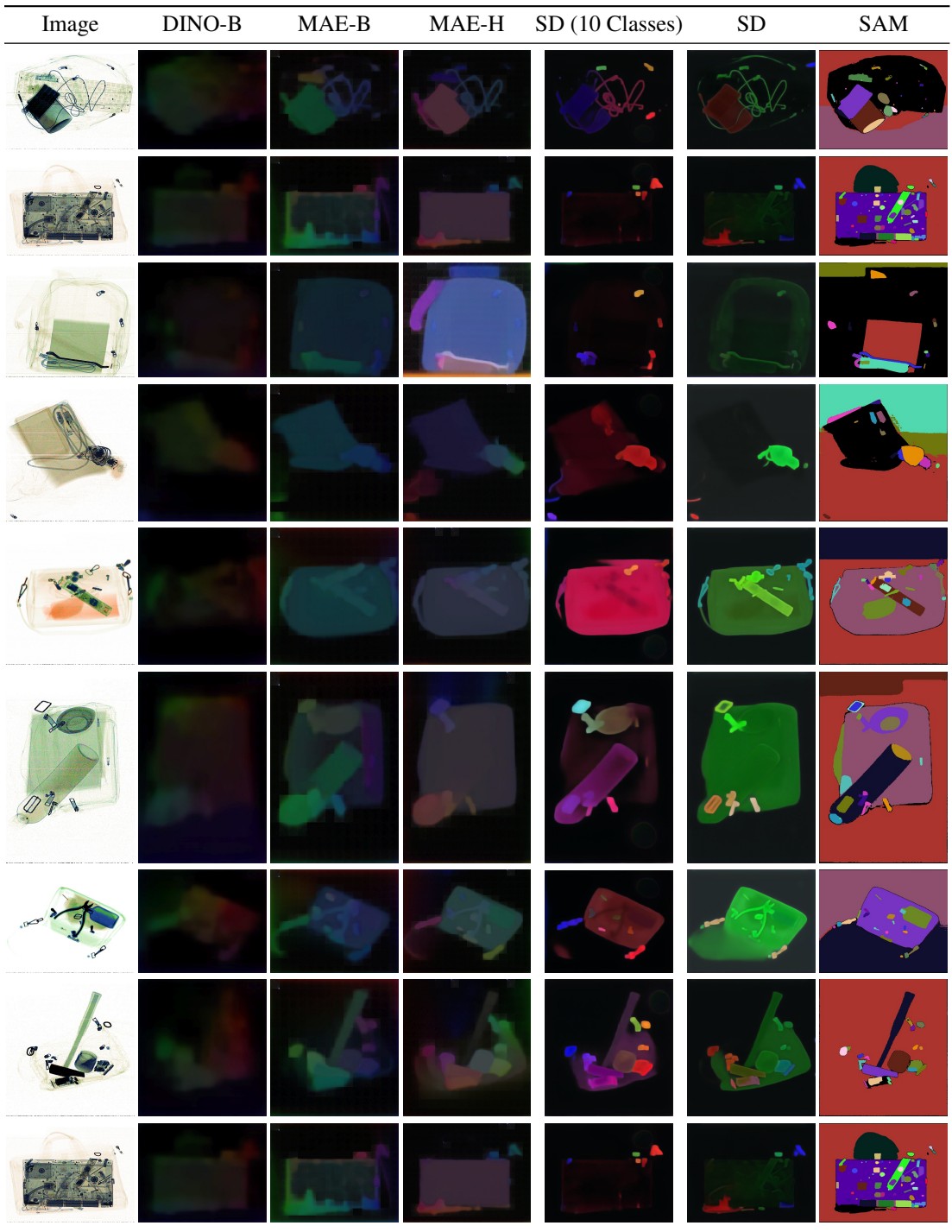

Figure 14: Qualitative Results on the PIDRay (Zhang et al., 2022a) dataset. These results are randomly chosen and not cherry-picked.

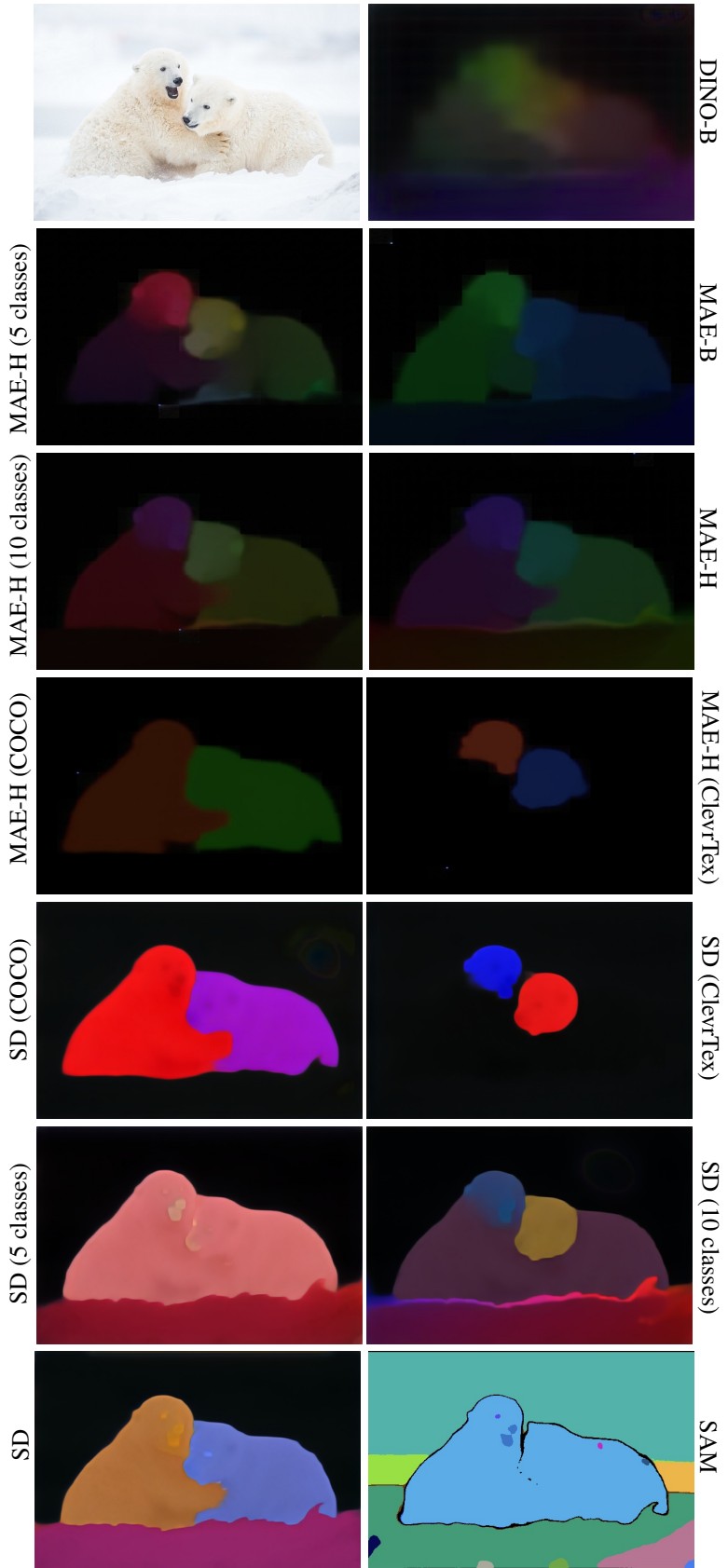

Figure 15: Qualitative comparison of all models on a challenging, in-the-wild image.

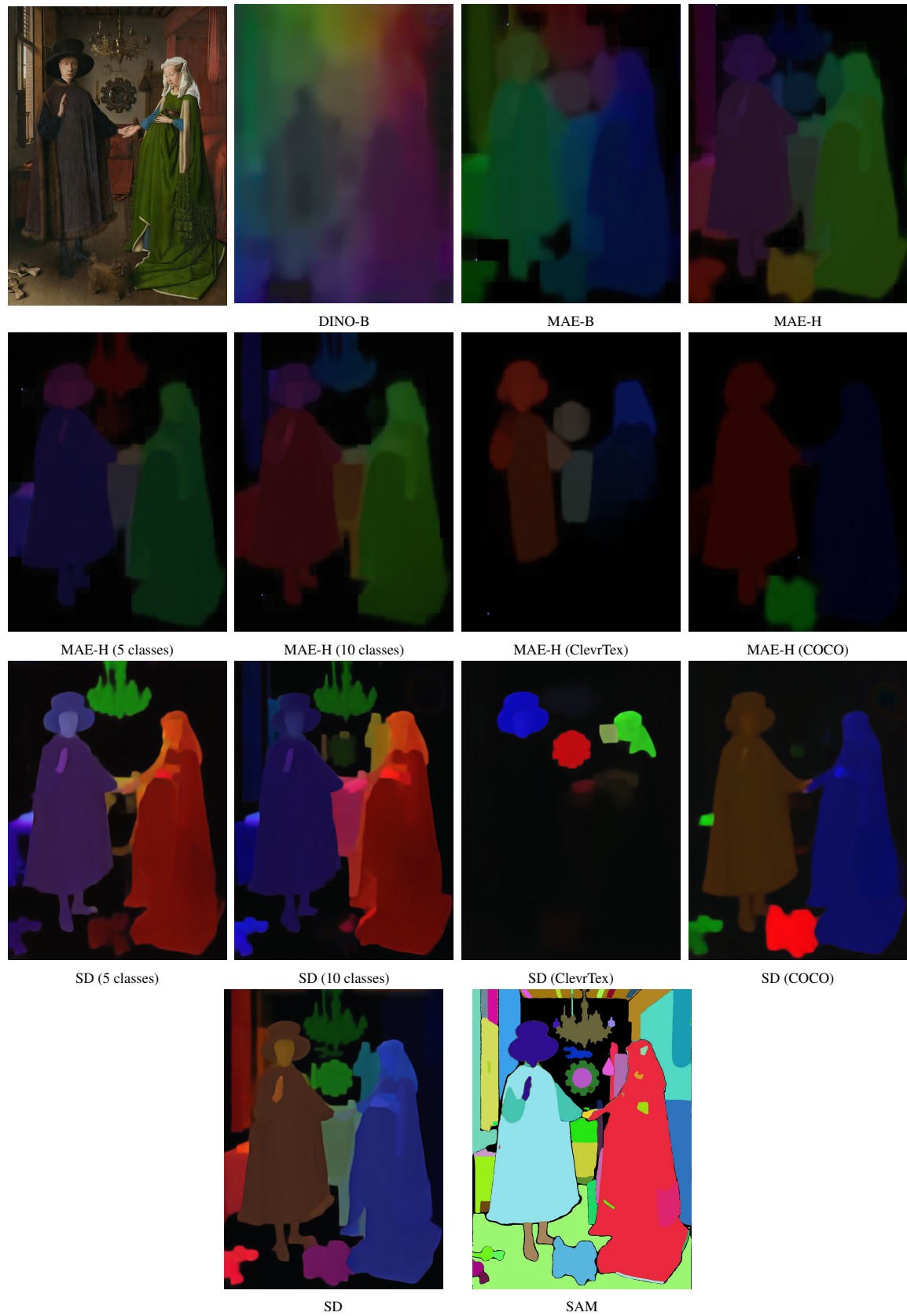

Figure 16: Qualitative comparison of all models on a challenging, in-the-wild image.

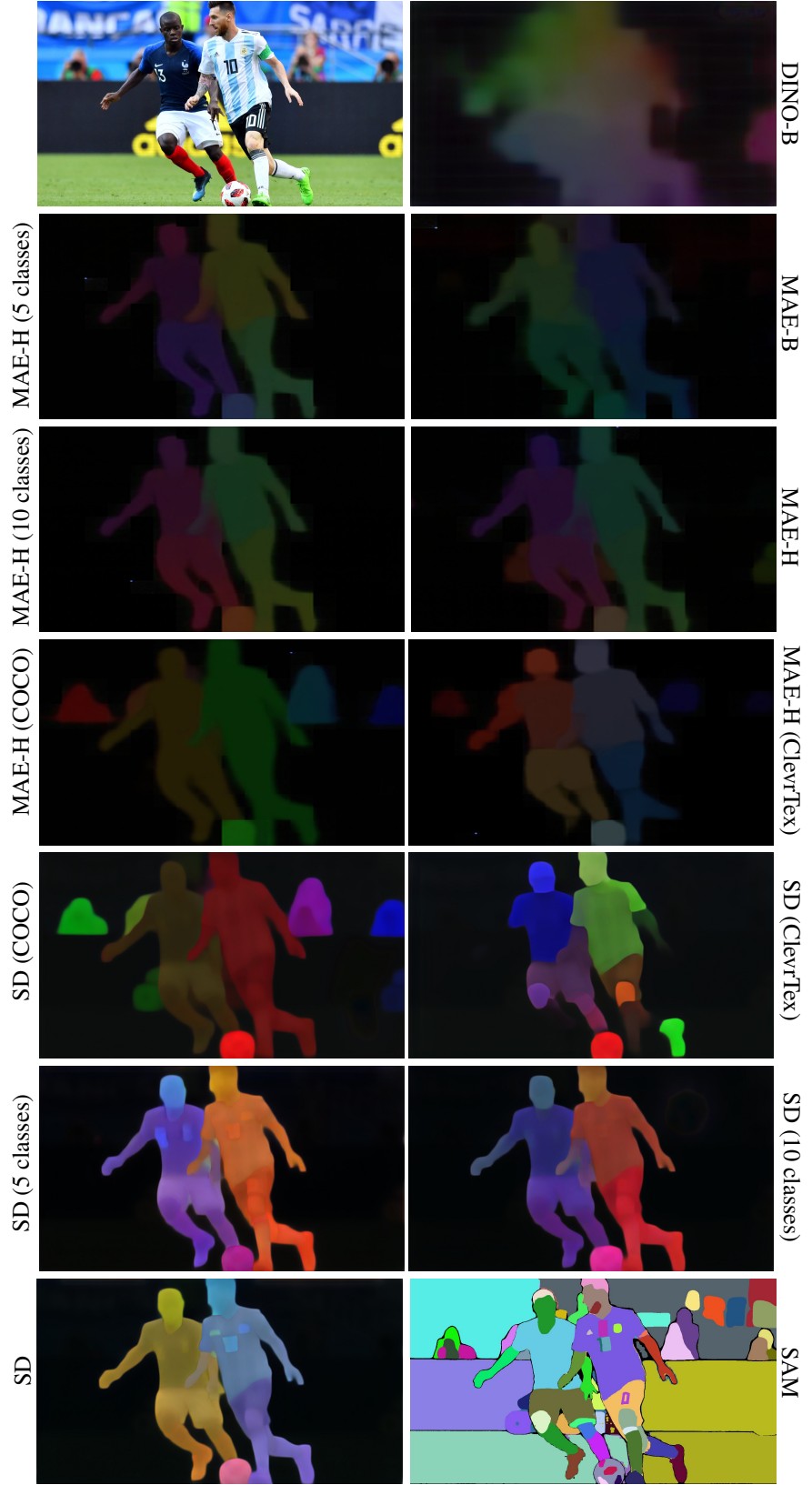

Figure 17: Qualitative comparison of all models on a challenging, in-the-wild image.

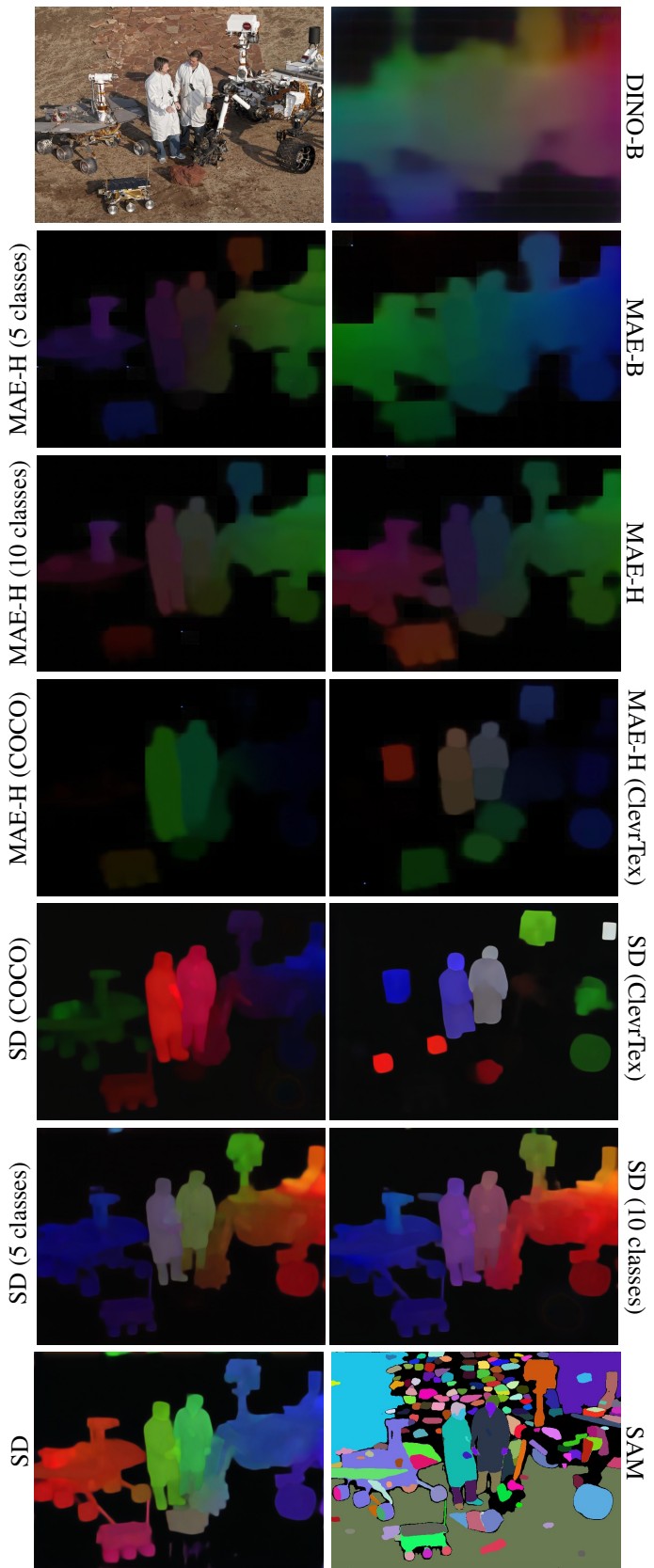

Figure 18: Qualitative comparison of all models on a challenging, in-the-wild image.

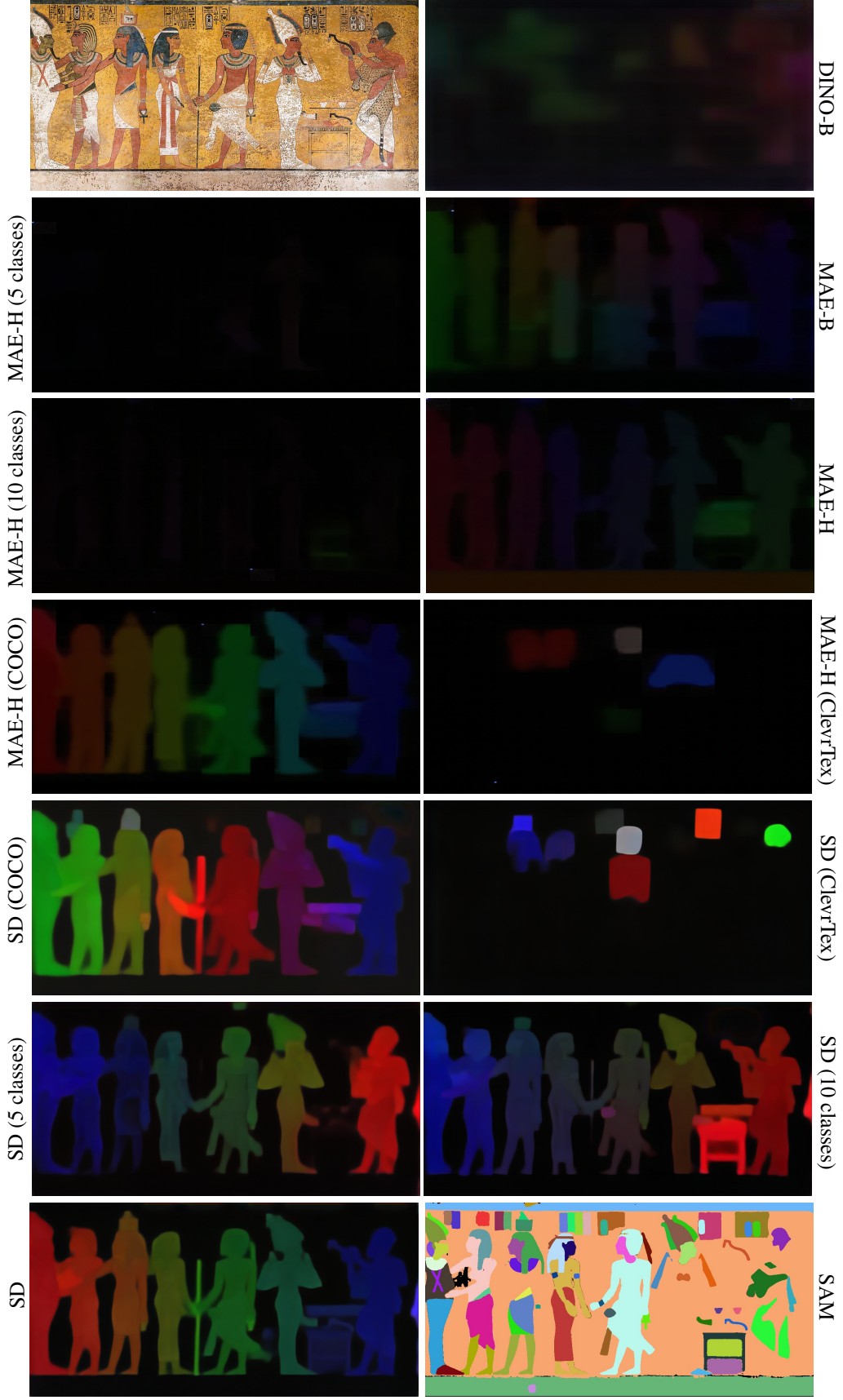

Figure 19: Qualitative comparison of all models on a challenging, in-the-wild image.

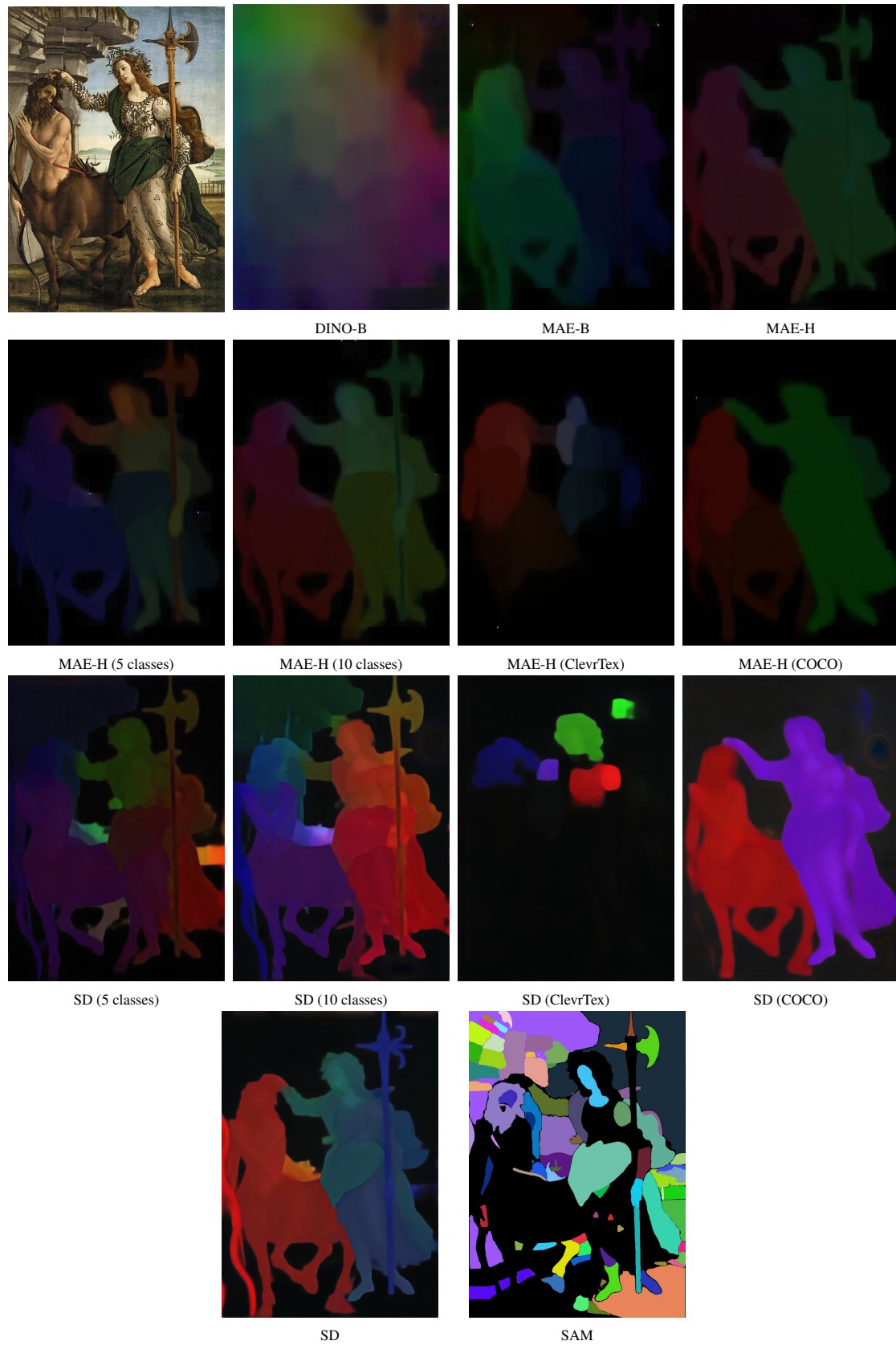

Figure 20: Qualitative comparison of all models on a challenging, in-the-wild image.

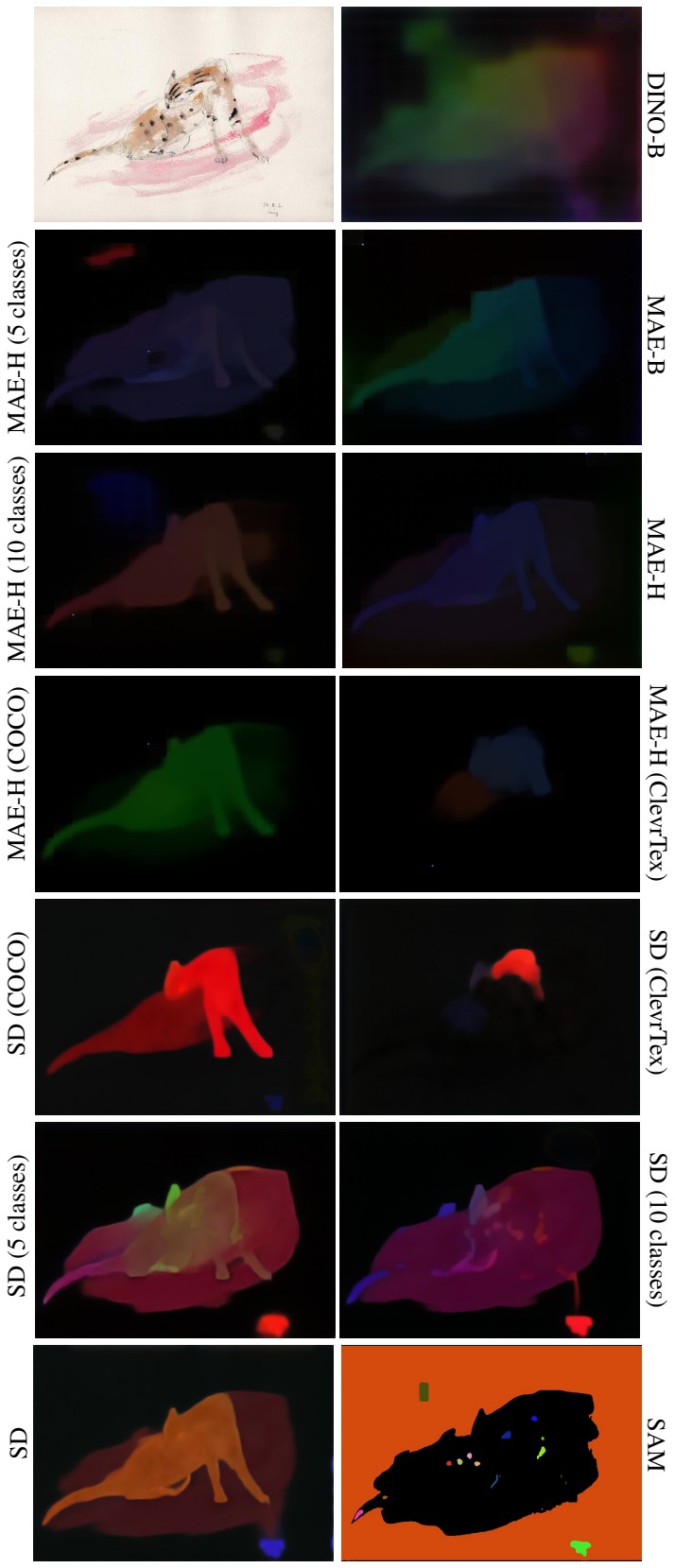

Figure 21: Qualitative comparison of all models on a challenging, in-the-wild image.

