# OpenReview forum: "gen2seg: Generative Models Enable Generalizable Instance Segmentation"
_ICLR.cc/2026/Conference — ICLR 2026 Poster_

### Official Review · Reviewer_P9Gg · 2025-10-25

**Soundness:** 2
**Presentation:** 3
**Contribution:** 2
**Rating:** 2
**Confidence:** 3

**Summary:**

The paper introduces a novel fine-tuning approach that repurposes generative models such as Stable Diffusion and Masked Autoencoders  for category-agnostic instance segmentation. Despite being fine-tuned only on limited synthetic data (indoor furnishings and cars), these models exhibit strong zero-shot generalization, segmenting unseen object types and image styles -- often surpassing the heavily supervised Segment Anything Model.

**Strengths:**

- Demonstrates good performance on unseen object categories and visual domains despite fine-tuning on a very limited dataset.

- Shows that generative models (Stable Diffusion, MAE) encode transferable perceptual grouping abilities, enabling segmentation without large-scale supervision.

- Achieves competitive results to SAM with a lightweight fine-tuning procedure using fewer resources and annotated masks.

**Weaknesses:**

- Questionable evaluation protocols

Actually, there exists an established experimental protocol for evaluating click-based segmentation methods, that implies testing on GrabCut, Berkeley, DAVIS, SBD, PascalVOC benchmarks. E.g., SimpleClick referred in the proposed paper as a baseline, reports values obtained in this benchmark. But here, the proposed method is not tested in the same setup, instead, an entirely different experimental protocol is used. Namely, SimpleClick was trained on synthetic datasets with limited number of classes and scenarios, and tested far beyond classes used for training. In such an artificial scenario, its low accuracy, otherwise suspicious, can be indeed expected -- as well as superior performance of a generalist model. When tested in the normal evaluation protocol, SimpleClick yields decent performance, though.

- Using diffusion (and even SD!) for click-based segmentation is not novel

This idea was exploited in the early Conditional Diffusion Network (CDNet) [1] that dates back to 2021. In CDNet, global similarity of features from clicks and potential target regions was used in a diffusion process that spreaded the predicted logits of clicks within locally connected regions. The most recent M2N2 [2] utilizes self-attention maps produced by Stable Diffusion for unsupervised and training-free approach to instance segmentation. None of these works is cited or discussed, which might make a reader to overestimate the novelty of the proposed approach.

[1] Conditional Diffusion for Interactive Segmentation. Xi Chen et al. ICCV 2021.

[2] Repurposing Stable Diffusion Attention for Training-Free Unsupervised Interactive Segmentation. Markus Karmann and  Onay Urfalioglu. In CVPR 2025.

- Results are questionable

The method was evaluated against trainable SimpleClick, custom pipeline based on DINO and off-the-shelf SAM. SimpleClick were trained according to custom training procedure, which naturally inhibits concerns about whether there were properly trained -- or their potential was not unleashed due to suboptimal hyperparameters, shortened training schedule etc. DINO was again used as a part of a custom pipeline, so similar concerns are applicable. Results achieved with SD, are significantly inferior to those of SAM on medium-sized and small-sized objects in COCO, DRAM, EgoHOS and PIDRay, in other words, in almost all scenarios, which makes me doubt the practical value of the proposed approach. It seems that on real world applications, it barely could compete SAM.

**Questions:**

- Why not use the well known evaluation protocol for assessing quality of click (point-prompt) interactive segmentation?

- Why not compare to pretrained SimpleClick but implement a custom training scheme on synthetic datasets?

- Why SAM and SimpleClick are selected as beaselines? I believe there are some more powerful methods that have been released since; at least SAM2, if not digging deeper [3].

[3] AdaptiveClick: Clicks-aware Transformer with Adaptive Focal Loss for Interactive Image Segmentation. Jiacheng Lin et al. In TNNLS, 2024.

---

> ### Author Response · Authors · 2025-11-22
>
> Our main goal is to show that, when adapted with our finetuning method, generative models show several useful emerging properties: 1) zero-shot generalization to classes far outside the finetuning domain 2) high-quality segmentation of fine structures 3) very fine object edges. Additionally, we our findings extend beyond diffusion models to the broader class of image generative models by showing these findings on ImageNet-pretrained MAE as well. *See response to reviewer BBSU for more details.*
>
> **Evaluation protocol**
>
> We used the evaluation strategy from the SAM paper (see Table 1). Due to SAM’s popularity, we feel this is also valid. The evaluation datasets we use are also selected from SAM paper.
>
> Following reviewer's suggestion, SimpleClick evaluation protocol can be used in two different ways:
> (1) Use SimpleClick model already trained on COCO/LVIS and test on the 5 datasets: We argue that this evaluation is not appropriate for our paper since it does not show that instance segmentation generalizes to object classes not seen in finetuning. Note that the object classes in the test set are already presented or are very similar to object classes in COCO dataset.
>
> (2) Train SimpleClick on our limited dataset and test on the 5 datasets: This is appropriate as it evaluates generalization. We did the training already in the main paper and did test it using SAM-style evaluation (see Table 1). Following reviewer's suggestion, we report the results of the same pretrained model on SimpleClick's evaluation test set and compare with ours below.
>
> | Dataset   | Method      | NoC@80% | NoC@85% | NoC@90% |
> |-----------|------------:|--------:|--------:|--------:|
> | GrabCut   | SimpleClick |  12.54  |  13.44  |  14.80  |
> | GrabCut   | MAE-B       |   4.70  |   5.92  |  11.22  |
> | GrabCut   | MAE-H       |   3.28  |   4.24  |   8.50  |
> | GrabCut   | SD          | **2.46**| **3.18**| **6.38**|
> | Berkeley  | SimpleClick |  13.42  |  14.65  |  16.18  |
> | Berkeley  | MAE-B       |   6.91  |   9.11  |  12.53  |
> | Berkeley  | MAE-H       |   3.81  |   6.11  |  10.81  |
> | Berkeley  | SD          | **2.86**| **4.58**| **8.11**|
> | DAVIS     | SimpleClick |  19.16  |  19.46  |  19.67  |
> | DAVIS     | MAE-B       |   8.48  |  12.71  |  15.54  |
> | DAVIS     | MAE-H       |   6.94  |  11.51  |  12.22  |
> | DAVIS     | SD          | **5.21**| **8.63**| **9.17**|
> | PascalVOC | SimpleClick |  12.95  |  13.77  |  14.75  |
> | PascalVOC | MAE-B       |   6.00  |   7.29  |  13.23  |
> | PascalVOC | MAE-H       |   4.51  |   6.33  |  11.65  |
> | PascalVOC | SD          | **3.38**| **4.75**| **8.74**|
> | SBD       | SimpleClick |  14.30  |  15.79  |  17.65  |
> | SBD       | MAE-B       |   7.44  |  11.05  |  13.96  |
> | SBD       | MAE-H       |   5.60  |   9.84  |  13.24  |
> | SBD       | SD          | **4.20**| **7.38**| **9.93**|
>
> Note that SimpleClick results on this setting are much worse than regular setting where it is trained on COCO/LVIS. We believe this happens since SimpleClick is designed to excel at the classes it is trained on rather than generalize.
>
> **Using other baselines like SAM2**
>
> As stated earlier, our goal is to show generalization rather than to produce SOTA results. Hence, we do not think evaluation on the most SOTA instance segmentation model that does not show generalization contributes to our paper. If there is any paper after SAM that shows better generalization, we should compare with it, but we have not found any such work. For instance, SAM2 is better than SAM but the improvement is mainly due to larger training set and video supervision which does not contribute to our goal of generalization. SAM2 performs the same as SAM on images when the finetuning data is the same (see SAM2 Tech report, Table 5).
>
>
> **Using diffusion (and even SD!) for click-based segmentation is not novel**
>
> We want to emphasize that we do not use a diffusion process in any of our models. Rather, we train a one step, deterministic image segmenter (Lines 357-360) from a generative model. Thus, CDNet is not very related to our work. We will clarify this in our paper. Similarly, we cited several related works that adopt diffusion models for segmentation (Lines 140-146) including M2N2 (Lines 141-142, 570-571) in the original submission and do not claim to be the first to use generative models for segmentation.
>
> **Results are questionable**
>
> We used the default hyperparameters for SimpleClick. We felt the current evaluation was the fairest possible way that we could think of to evaluate DINO. We are happy to run additional experiments if you have any suggestions. However, our key findings do not rely on the validity of this baseline, as we are showcasing interesting findings that emerge when finetuning generative models for instance segmentation.

---

> > ### Author Response · Authors · 2025-11-22
> >
> > **Results are inferior to SAM**
> >
> > We feel the review oversimplifies our findings. First, given that our method has never seen masks of objects in DRAM, EgoHOS, or PIDRay, it is remarkable that our model can recover 96%, 71%, and 70% of SAMs' accuracy on the respective datasets regardless of object size. This shows our method's extreme zero-shot generalization. Second, the review ignores that we outperform SAM on iShape (fine structures) by over 305%. Like every method, we have advantages and disadvantages; our advantage is fine structures but our disadvantage is small objects. We hope that future work will combine our findings with SAM to build models that utilize both sides' strengths.
> >
> > **Practical impact**
> >
> > We disagree that our findings do not have practical impact. For example, our model’s edges are more precise compared to SAM’s, and the fact that this persists even when finetuned on polygonal edges may be of use to someone without detailed labels who want precise masks. Similarly, our model substantially outperforms SAM in segmenting fine structures. A robot that wants to separate entangled shoelaces would be able to use our model off-the-shelf but not SAM off-the-shelf.

---

> ### Comment · Reviewer_P9Gg · 2025-11-26
>
> I thank authors for the rebuttal.
>
> Authors confirm that
>  - their approach is far behind state-of-the-art in a well-know click-based instance segmentation task;
>  - they are not the first to train instance segmentation on top of pre-trained generative model;
>  - there exist even training-free approaches to do it.
>
> Based on this, I don't see reasonable novelty, and keep my original score.

---

> > ### Author Response · Authors · 2025-12-04
> >
> > We disagree.
> > 1. Our goal is not to achieve SOTA numbers for segmentation. Our goal is to show that generative models can be finetuned using masks with a very limited set of categories (only five categories in some experiments) and generalize to produce masks of several other categories in zero-shot setting as the output of the model.
> >
> > 2. As cited in the paper, there is prior work finetuning generative models for segmentation, but they do not show generalization with limited finetuning data.
> >
> > 3. We agree that there are papers showing one can do segmentation without training by tapping into the attention map of the generative models. However, our finding is different since we show the masks can be generated as colored image output (in an image-to-image translation setting) with high quality and rich generalization.

---

### Official Review · Reviewer_BBSU · 2025-10-26

**Soundness:** 2
**Presentation:** 3
**Contribution:** 2
**Rating:** 4
**Confidence:** 4

**Summary:**

The paper explores how generative models pretrained for image synthesis can be repurposed for perceptual organization. By finetuning Stable Diffusion and MAE with an instance coloring loss on limited object types, the models achieve strong zero-shot generalization, segmenting unseen objects with high accuracy. Remarkably, they rival or even surpass SAM on fine structures, demonstrating that generative models inherently learn transferable object grouping mechanisms without large-scale supervision.

**Strengths:**

1. GenSeg demonstrates that simply finetuning generative models such as Stable Diffusion can yield strong performance in category-agnostic instance segmentation.

2. The paper is clearly written and easy to follow.

3. GenSeg achieves SAM-level performance in certain out-of-distribution segmentation scenarios.

**Weaknesses:**

1. The novelty of this work is limited. Previous studies [1][2][3] have already shown that generative models can exhibit open-set segmentation capabilities without task-specific training. This paper mainly extends those ideas to category-agnostic instance segmentation.

2. Methodologically, the paper lacks innovation. Finetuning generative models for segmentation is a well-established approach, and the proposed loss function is highly similar to existing techniques [4][5].

[1] Generative prompt model for weakly supervised object localization[C]//Proceedings of the IEEE/CVF International Conference on Computer Vision. 2023: 6351-6361.

[2] Diffumask: Synthesizing images with pixel-level annotations for semantic segmentation using diffusion models[C]//Proceedings of the IEEE/CVF International Conference on Computer Vision. 2023: 1206-1217.

[3] Open-vocabulary panoptic segmentation with text-to-image diffusion models[C]//Proceedings of the IEEE/CVF conference on computer vision and pattern recognition. 2023: 2955-2966.

[4] Recurrent pixel embedding for instance grouping[C]//Proceedings of the IEEE conference on computer vision and pattern recognition. 2018: 9018-9028.

[5] Efficient and accurate arbitrary-shaped text detection with pixel aggregation network[C]//Proceedings of the IEEE/CVF international conference on computer vision. 2019: 8440-8449

**Questions:**

The major weakness of the paper is limited novelty.

---

> ### Author Response · Authors · 2025-11-22
>
> Thanks for your feedback. We appreciate you found our paper “clearly written and easy to follow,” and liked our model’s generalization.
>
> We do not claim a novel loss. **The novelty lies in the discovery of several emergent phenomena after finetuning, listed below.** We intentionally intended to keep the finetuning loss as simple and intuitive as possible, and highlight that it is inspired by [4] and [5] (see lines 226-227). Similarly, we do not claim to be the first to use generative models for instance segmentation (lines 140-146).
>
> *We have summarized these emergent properties (our “novel” findings) below:*
>
> 1. Our models finetuned for instance segmentation generalize to classes they have never seen masks of in finetuning. We show this emerges even when only 5 object types are present in the finetuning set (Table 2). This suggests that once the generative model has learned the notion of an instance mask, it transfers across object types. This is different from prior works [1, 2, 3], which utilize the prior of a diffusion model, but only test on classes they have trained the model on.
>
> 2. Our models excel at segmenting fine structures such as shoelaces, wires, or fences. Specifically, our best models outperform SAM on the iShape dataset by over 305%. This quality persists even when the finetuning set does not contain masks of fine objects (i.e. COCO or ClevrTex, Table 2).
>
> 3. Our models produce extremely fine mask edges (Table 3). This persists even when the finetuning dataset contains mostly polygonal edges (COCO), suggesting generative models learn a robust internal representation of object boundaries.
>
> 4. We extend our findings beyond diffusion models to MAE models (encoder+decoder). To the best of our knowledge, we are the first to do this. We build on this by showing that the core instance segmentation knowledge resides in the last layers of the decoder, closest to where image synthesis occurs (Table 7). In particular, our findings contradict the conventional practice of discarding the decoder after pretraining MAE.
>
> **Our work builds on these principles to showcase “emergent” advantages generative model-based instance segmenters have over current models. To restate: the findings of these “emergent” properties is the novelty, not the finetuning method.**
>
> To elaborate, we use the finetuning style because 1) it introduces no additional parameters and 2) the method does not have any complicated bells-and-whistles 3) it allows to repurpose the image-level output at the original resolution, and not low-resolution internal features (as prior works have done). This makes it easy to understand and helps us minimizes architectural changes, isolating contribution from the generative model.

---

> ### Comment · Reviewer_BBSU · 2025-11-23
>
> Thank you for your response.
>
> 1. Regarding your point about ``*This suggests that once the generative model has learned the notion of an instance mask, it transfers across object types.*'', I do not consider this a meaningful innovation. Prior works such as [1] and [2] have already shown that diffusion models can achieve open-set segmentation without any mask annotations. Fine-tuning the model in this paper with a small number of mask labels only modestly extends this existing capability—for example, by improving boundary segmentation—but does not introduce a substantive conceptual advancement.
>
> 2. I acknowledge the paper’s empirical contribution; the final model demonstrates strong generalization performance. However, the claim that generative models ``*learn a robust internal representation of object boundaries*'' has already been validated by [1] and [2].
>
> 3. Extending diffusion models to MAE models does not constitute a major contribution. Both diffusion models and MAE models are generative models, and their unsupervised segmentation performance has been extensively explored. In addition, several prior works have emphasized the importance of the MAE decoder for dense prediction tasks, such as [3]. Therefore, ``*our findings contradict the conventional practice of discarding the decoder after pretraining MAE*'' does not substantiate a unique contribution of this paper.
>
> Based on the above reasons, I will keep my rating at this stage.
>
> [1] Tian, Junjiao, et al. "Diffuse attend and segment: Unsupervised zero-shot segmentation using stable diffusion." CVPR, 2024.
>
> [2] Wang, Jinglong, et al. "Diffusion model is secretly a training-free open vocabulary semantic segmenter." IEEE TIP, 2025.
>
> [3] Liu, Feng, et al. "Integrally migrating pre-trained transformer encoder-decoders for visual object detection." ICCV, 2023.

---

> > ### Author Response · Authors · 2025-11-26
> >
> > Thank you for your response.
> >
> > 1. First, as shown in the qualitative figures and introduction of [1] and [2], the masks derived from attention maps are highly semantic (Figure 2 in [1], title of [2]). Ours is different since we focus on "instance level" segmentation masks rather than category level. Additonally, the mask is the output of the model itself (image-to-image translation) rather than a postprocess of the attention maps of the model like in [1] and [2]. Also, [2] uses cross attention from text conditioning while our model can run on image-only models too. We will clarify in the writing.
> >
> > 2. Additionally, we believe a core conceptual advancement is that in some cases, a generative prior is better than SAM. Specifically, neither [1] nor [2] highlight generative models' superiority on fine structures or precise edges relative to existing models such as SAM. By "robustness" in the quoted sentence, we meant robustness to noisy and contradictory annotations at the finetuning stage (e.g, polygonal contours of COCO dataset). This cannot be shown in [1] and [2] since they do not do finetuning. We hope to encourage future work to build on the strengths of both generative and discriminative models.
> >
> > 3. We were not aware of [3] and will cite it and modify the main contribution reflecting on this. Note that [3] focuses mainly on object detection and has very limited experiments on instance segmentation using a Mask RCNN head. Also, this does not degrade our main contribution since [3] focuses on a highly supervised regime and does not show generalization for unseen categories or detailed segmentation.

---

### Official Review · Reviewer_XWuX · 2025-10-31

**Soundness:** 4
**Presentation:** 4
**Contribution:** 4
**Rating:** 8
**Confidence:** 3

**Summary:**

This work demonstrates that generative models such as Stable Diffusion and Masked Autoencoders (MAE) can be fine-tuned on a small set of labeled object masks to achieve general-purpose instance segmentation.
Remarkably, the fine-tuned models exhibit strong zero-shot generalization, accurately segmenting objects of entirely unseen types and styles.
They also produce sharper boundaries and handle fine-grained structures with notable precision, behaviors that likely stem from their generative pretraining.
These results suggest that generative pretraining inherently encodes a grouping and compositional understanding of the visual world, enabling segmentation capabilities that are more generalizable and human-like across diverse domains.

**Strengths:**

### 1. Surprising zero-shot generalization and conceptual novelty:
It is impressive that the proposed generative models exhibit strong zero-shot generalization, accurately segmenting objects of entirely unseen types and styles.
This suggests that generative pretraining inherently encodes a grouping mechanism that transfers across categories and domains, supporting more generalizable and human-like perception.
This is arguably the first convincing demonstration of the potential of generative segmentation models to surpass traditional discriminative approaches in terms of generalization capability.

### 2. Robustness to visual perturbations:
The models show high robustness to variations in color, illumination, and texture, indicating that their segmentation ability does not rely merely on low-level visual cues.
This robustness highlights a deeper structural or semantic understanding learned through generative pretraining.

### 3. Strong qualitative performance:
Qualitative visualizations (see pages 7–9) show coherent reconstructions and smooth, well-aligned segmentation boundaries.
The results are visually compelling and provide strong evidence for the model’s superior perceptual grouping and boundary accuracy compared to discriminative baselines.

**Weaknesses:**

### 1. Limited performance on small objects:
As shown in Table 1, the proposed approach performs well on large objects but struggles significantly with small-object segmentation.
This limitation suggests that the model’s generative priors may bias it toward capturing dominant, large-scale structures, leaving fine-scale, small instances underrepresented.

### 2. Imprecise instance coloring and unclear mask extraction details:
The color-based instance representation occasionally produces noisy or “dirty” colors, which raises concerns about the consistency and robustness of the predicted masks.
The paper would benefit from a clearer explanation of how these colored outputs are converted into binary or instance masks, with more explicit technical detail in the Methods section.

### 3. Limited extensibility to semantic or text-guided segmentation:
Since the framework relies heavily on color-based encoding, its applicability to semantic or  text-guided segmentation appears constrained.
The authors may consider discussing how this approach could be extended to tasks requiring semantic consistency or text/image guidance, which could broaden the impact of the work in future research.

**Questions:**

The main concern lies in the model’s difficulty with small-object segmentation.
Generative segmentation models, while powerful, can be somewhat uncontrollable, as it is often unclear what structures or priors the model has actually learned.
This may explain why small or less prominent objects tend to be omitted in the predictions.

Could the authors elaborate on how controllability might be improved in such generative segmentation frameworks?
Additionally, what potential strategies do you envision for enhancing the detection and segmentation of small objects within this generative paradigm?

---

> ### Author Response · Authors · 2025-11-22
>
> Thank you very much for your feedback. We greatly appreciate you found our findings the "first convincing demonstration" to surpass existing models in zero-shot generalization. We are also glad you found our examples "visually compelling."
>
> **Controllability in generative models**
>
> We agree on the controllability issue of generative models. Currently, the base models we finetuned themselves are not very good at controlling small objects. For example, it is well known that Stable Diffusion will overfocus on a single object or neglect multiple objects in a prompt [R21].
>
> While there are some recent models (i.e. FLUX, Nano Banana, Reve) that have improved substantially in editing or controlling small objects, they are either private or far too large to fit on 48GB GPUs. We believe that the segmentation quality of small objects will improve with larger models that are better able to control small objects in generation.
>
> **Instance coloring to mask**
>
> Our feature-to-mask method is inspired by attention. We first calculate a "query" feature by taking the prompt-pixel's respective feature, and average it with some nearby pixels. We then compute euclidean distance between all the pixels and the query to produce a similarity map. To reduce "noisiness" in masks we perform joint bilateral filtering to smooth out the similarity map. We then normalize between 0-255 and threshold it with a fixed value (for most experiments, 3) such that values above it are considered the mask.  Please let us know if you have any additional questions.
>
> **Improving small objects in future work**
>
> We hypothesize that a future segmentation model will not be solely discriminative nor generative, but building on the unique strengths and advantages of both methods. We leave this investigation to future work.
>
> [R21] Hila Chefer*, Yuval Alaluf*, Yael Vinker, Lior Wolf, Daniel Cohen-Or. Attend-and-Excite: Attention-Based Semantic Guidance for Text-to-Image Diffusion Models. In SIGGRAPH 2023.

---

### Official Review · Reviewer_qeaH · 2025-11-01

**Soundness:** 3
**Presentation:** 3
**Contribution:** 3
**Rating:** 6
**Confidence:** 4

**Summary:**

The authors propose a method to adapt image generation models to perform instance segmentations across diverse classes including those unseen during training. Impressive improvements over strong baselines have been shown in the paper. Image generation models are adapted to output segmentation maps in a single step by training on various image segmentation datasets with novel loss functions. Ablations and analysis is also included.

**Strengths:**

The performance gains are good over baselines trained on the same data. The authors also test on unseen domains such as luggage X-ray images where they observe similar trends. Qualitative comparison is very helpful to understand these claims and quite impressive. The fact that the model performs so well on unseen domains makes the contribution of the novel losses and method design strong.

**Weaknesses:**

According to evaluation, the Stable Diffusion variant of their model is performing the best. However this a old model and it would be good to see how the performance scales when used with recent DiT based models such as SD3 and Flux.

The authors hint at part level understanding in Figure 3. It would good to perform evaluation on some part segmentation datasets [1,2] to check generalization of these models to finer granularity levels.

Training is conducted on a relatively small and limited diversity dataset. It would be good to explore the effect of expanding the dataset, eg. using SA-1B (or a subset).


[1] Saha, O., Cheng, Z. and Maji, S., 2022, October. Improving few-shot part segmentation using coarse supervision. In European Conference on Computer Vision (pp. 283-299). Cham: Springer Nature Switzerland.

[2] Vedaldi, A., Mahendran, S., Tsogkas, S., Maji, S., Girshick, R., Kannala, J., Rahtu, E., Kokkinos, I., Blaschko, M.B., Weiss, D. and Taskar, B., 2014. Understanding objects in detail with fine-grained attributes. In Proceedings of the IEEE conference on computer vision and pattern recognition (pp. 3622-3629).

**Questions:**

Will the pretrained models also generalize to medical segmentation scenarios?

---

> ### Author Response · Authors · 2025-11-22
>
> We thank the reviewer for the positive feedback, especially noting that “the improvements are quite impressive over strong baselines.” We appreciate the recognition that our qualitative comparisons were “very helpful to understand these claims and quite impressive.”
>
>
> **Recent diffusion models**
>
> It is challenging to finetune larger models as they don't fit on 48GB GPUs which are the only ones available to us. We are looking into alternative options and if possible, we will add it to camera-ready.
>
> **Part segmentation datasets**
>
>
> |Class|#Parts|No Compositionality (IoU)|MAE-H (IoU)|SD (IoU)|SAM (IoU)|
> |---|---|---|---|---|---|
> |aeroplane|19|17.5|10.3|19.7|48.6|
> |bird|13|15.6|14.2|20.3|36.3|
> |bicycle|6|19.5|**28.4**|**27.6**|46.6|
> |bottle*|2|53.4|28.0|34.4|53.1|
> |bus|46|12.7|7.5|9.5|56.0|
> |car*|30|18.7|10.6|13.6|57.7|
> |cat|17|10.7|**22.5**|**29.0**|39.1|
> |cow|19|11.4|11.0|**17.3**|26.9|
> |dog|18|11.0|**19.0**|**25.9**|34.2|
> |horse|21|9.2|10.7|**17.8**|26.2|
> |motorbike|7|15.5|**21.1**|**29.1**|54.0|
> |person|24|9.2|**27.3**|**31.0**|54.1|
> |pottedplant*|2|54.4|36.2|38.4|58.4|
> |sheep|19|14.4|10.0|15.5|27.1|
> |train|52|30.9|25.0|34.0|38.7|
> |tvmonitor*|1|55.5|51.3|57.4|66.9|
>
>
> Thanks for suggesting experiments with part segmentation datasets. We hope this will provide additional insight to future readers.
>
> We experiment with the PASCAL-Part [R11] dataset, which annotates PASCAL-VOC with part level annotations. This allows us to investigate a diverse range of object types for part-level groupings. The bolded values represent classes for MAE-H or SD where our model performed at least 5 points better than the baseline. The baseline "No Compositionality" is calculated by evaluating IoU between every part mask and its whole object mask (i.e. IoU of cat head vs whole cat). This represents the values close to what we would get if there was no part-level grouping at all. We also report results from SAM to serve as a high-water mark and provide context for our results.
>
> In this table, we do see signs of part-level grouping for some classes such as bike, cat, cow, dog, horse, motorcycle, and person. This emerges without any supervision on part or whole-object masks of these categories, suggesting it is inherent to the generative model. Classes marked with "\*" are included in our Hypersim+VK2 mix.
>
> **Limited diversity dataset**
>
> We have reported results for finetuning on COCO dataset in Table 2 of our original submission which has much higher diversity than our limited set. However, we intentionally use a finetuing dataset with limited diversity since our goal is to evaluate the generalization of generative models in generating masks of unseen categories.
>
> **Medical segmentation scenarios**
>
> |SimpleClick| MAE-H | SD | SAM
> |:--------: | --------- | :--------: |:--------: |
> |16.1 | 41.2 | 44.5 | 74.8
>
> We evaluate on the BBBC039 dataset [R12], which is focused on flourescent microscopy of cell nuclei. Our models perform decently, suggesting there is some zero-shot generalization to medical scenarios. However, more challenging scenarios such as tumor or angiograph segmentation are challenging for humans without medical experience as well. Following the same trend of the paper, our method outperforms SimpleClick which is a discriminative baseline trained on the same masks.
>
> [R11] Xianjie Chen, Roozbeh Mottaghi, Xiaobai Liu, Sanja Fidler, Raquel Urtasun, Alan Yuille. Detect What You Can: Detecting and Representing Objects using Holistic Models and Body Parts. CVPR 2014.
>
> [R12] Vebjorn Ljosa, Katherine L Sokolnicki & Anne E Carpenter. Annotated high-throughput microscopy image sets for validation. Nature Methods, 2012.

---

### Author Response · Authors · 2025-12-04

We would like to thank all reviewers; the suggestions have improved our paper.

**Novelty**
Our core novelty is the findings of the emergent properties such as (1) zero-shot generalization, (2) superior performance on mask boundaries (even with polygonal finetuning masks), and (3) detailed fine structure segmentation (200+% relative to SAM). In contrast, the works cited by reviewers BBSU and P9Gg are focused on showing the presence of segmentation in attention maps and do not highlight any such generalization or superior performance.

**Experimental Protocol**
We appreciate that reviewers qeaH, XWuX, and BBSU highlighted our "strong" empirical results. Specifically, our models perform very close to SAM in a diverse range of settings. The experimental protocol suggested by Reviewer P9Gg, which is in SimpleClick (Liu et al., 2023a), is used to evaluate mask accuracy when the train and test sets contain very similar categories. Hence, our results are lower than SOTA since we evaluate our model in a generalization setting where train and test categories are very different from each other.

---

### Meta-Review · Area_Chair_MnbP · 2025-12-26

**Summary:**

This paper offers an interesting angle on the question whether visual generative models perform visual grouping, usually considered a cornerstone of the human visual system: the authors fine-tune an image diffusion model (and an MAE model) to produce instance segmentation masks (colored pixels for each instance) on a small set of indoor scenes and/or street environments and demonstrate reasonable transfer to unseen classes, in some cases from only 5 annotated object categories during fine-tuning.

Even though the paper does not constitute a state of the art in transfer learning or few-shot segmentation, it provides a novel angle on how visual generative models can be shown to excel at a task that requires visual grouping.

There is extensive literature on image segmentation with generative models (incl. diffusion models). Reviewers highlighted that some of this work was not sufficiently discussed or contextualized. The authors have made a good attempt at addressing this during the rebuttal and promised to update the paper with more extensive discussion and more precise claims around novelty.

Given that the paper is otherwise of high quality, and the finding is interesting to the community, the AC believes that it can be accepted as the authors can be expected to address the main remaining reviewer concerns with minor revisions of the paper.

**Reviewer Concerns:**

The main concerns by the reviewers revolve around the novelty of the finding: there is existing literature on evaluating the segmentation capabilities (sometimes zero-shot) of diffusion models, from which a similar conclusion can be drawn, i.e. visual generative models trained on large-scale image data perform or significantly facilitate visual grouping. This is a significant concern that has only been partially addressed in the rebuttal: the authors clarify that existing works focus on other aspects of segmentation generalization, e.g. emergent segmentation via attention maps, but none cover the precise setting of this paper. While this isn’t a strong argument for novelty, the paper does add value to the existing literature of evaluating generative models for their segmentation capabilities and has an interesting finding.

Another concern, namely that the results lack significantly behind state-of-the-art models in instance segmentation, has been commented on in the rebuttal: the authors clarified that state-of-the-art instance segmentation performance is not a claim made in the paper and not the reason why their finding is of interest to the community. The model does show positive results on generalizing to broad unseen classes from just a few training classes (in some cases as few as 5), which is a valuable and interesting finding, even though results are far behind state of the art generalist instance classification models in most tested cases.

**Reviewer Scores:**

None of the reviewers would have likely changed their score, putting this paper right on the borderline.

The AC weighed all comments and concerns and recommends acceptance. The main concerns around novelty and lack of practical state-of-the-art results can be addressed by adequate positioning of the paper, given that the finding does have limited, but still sufficient novelty, the positioning of the paper does not rely on SotA results but rather on a detailed analysis of a form of generalization that in this precise form has not been reported before and is likely of interest to the community.

---

### Decision · Program_Chairs · 2026-01-26

Accept (Poster)